# SAMPLE-EFFICIENT EVIDENCE ESTIMATION OF SCORE BASED PRIORS FOR MODEL SELECTION

**Frederic Wang & Katherine L. Bouman**
Department of Computing and Mathematical Sciences
Caltech
Pasadena, CA 91125, USA
{fzwang, klbouman}@caltech.edu

## ABSTRACT

The choice of prior is central to solving ill-posed imaging inverse problems, making it essential to select one consistent with the measurements $\boldsymbol{y}$ to avoid severe bias. In Bayesian inverse problems, this could be achieved by evaluating the model evidence $p(\boldsymbol{y} \mid M)$ under different models $M$ that specify the prior and then selecting the one with the highest value. Diffusion models are the state-of-the-art approach to solving inverse problems with a data-driven prior; however, directly computing the model evidence with respect to a diffusion prior is intractable. Furthermore, most existing model evidence estimators require either many pointwise evaluations of the unnormalized prior density or an accurate clean prior score. We propose `DiME`, an estimator of the model evidence of a diffusion prior by integrating over the time-marginals of posterior sampling methods. Our method leverages the large amount of intermediate samples naturally obtained during the reverse diffusion sampling process to obtain an accurate estimation of the model evidence using only a handful of posterior samples (e.g., 20). We also demonstrate how to implement our estimator in tandem with recent diffusion posterior sampling methods. Empirically, our estimator matches the model evidence when it can be computed analytically, and it is able to both select the correct diffusion model prior and diagnose prior misfit under different highly ill-conditioned, non-linear inverse problems, including a real-world black hole imaging problem.

## 1 INTRODUCTION

In Bayesian inverse imaging problems across science and engineering, the chosen prior distribution $p(\boldsymbol{x})$ plays a pivotal role in shaping the posterior $p(\boldsymbol{x} \mid \boldsymbol{y})$ from which candidate solutions are drawn (Feng et al., 2024; Stuart, 2010). In ill-posed settings, the prior acts as a regularizer, steering reconstructions toward solutions consistent with expected image properties. If the ground truth image lies outside the support of the chosen prior, the resulting reconstruction can be severely biased.

A central challenge in Bayesian inference is how to select a prior to use when the true distribution is unknown. Consequently, the priors used in practice are often chosen somewhat arbitrarily. If the likelihood of a prior could be easily evaluated, it would provide a principled basis for prior selection. Specifically, given a set of potential models $\{M_i\}$ that specify the prior $p(\boldsymbol{x} \mid M_i)$, we would like to find the most likely candidate model by computing $p(M_i \mid \boldsymbol{y}) \propto p(\boldsymbol{y} \mid M_i)p(M_i)$. It is commonly assumed that $p(M_i)$ is uniform, so it suffices to compute the *(log) model evidence*, $\log p(\boldsymbol{y} \mid M_i)$. Model evidence is not only useful for selecting the best prior for reconstruction, but also for quantifying epistemic uncertainty and assessing the assumptions underlying a model or physical theory (Trotta, 2007).

Although the model evidence provides a principled way to select the best prior for reconstruction, computing it generally requires evaluating an intractable integral over the full prior: $\log p(\boldsymbol{y} \mid M_i) = \log \int p(\boldsymbol{y} \mid \boldsymbol{x}, M_i)p(\boldsymbol{x} \mid M_i)d\boldsymbol{x}$. This motivates the development of estimation methods, such as sequential Monte Carlo (Del Moral et al., 2006; Chen et al., 2024), nested sampling (Skilling, 2006; Cai et al., 2022), annealed importance sampling Neal (2001); Doucet et al. (2022), and harmonic mean estimators (Newton & Raftery, 1994; McEwen et al., 2021; Polanska et al., 2023;

Spurio Mancini et al., 2023). However, these methods rely on sampling using the clean prior score $\nabla_{\boldsymbol{x}} \log p(\boldsymbol{x})$ or the unnormalized density $\log p(\boldsymbol{x})$, which may be inaccurate for out-of-distribution $\boldsymbol{x}$, expensive to evaluate, and can lead to slow mixing due to the ill-conditioned data geometry.

Bayesian inverse imaging solvers achieve state-of-the-art performance with diffusion-based priors (Song et al., 2020; Ho et al., 2020; Song et al., 2021b; Jalal et al., 2021; Mardani et al., 2023), yet an accurate model evidence remains intractable using existing methods. Diffusion models generate unconditional samples by denoising a sequence of intermediate noised images, each corresponding to a time-marginal distribution between pure noise and the clean image. Diffusion-based posterior sampling methods adhere to posterior time-marginals instead (Chung et al., 2022; Song et al., 2023; Coeurdoux et al., 2024; Li et al., 2024; Zhu et al., 2023; Wu et al., 2024; Zhang et al., 2025). While efficient diffusion density estimators have been proposed (Skreta et al., 2024), density-based estimation methods are still typically high-variance for high dimensional problems and require thousands of posterior samples, which is computationally challenging with diffusion models. Furthermore, as diffusion models learn the score of intermediate noised priors, their estimates of the clean prior score are usually less accurate and/or highly ill-conditioned, making it difficult for traditional methods to explore the full distribution, while using their score from a higher noise levels can add bias. As a result, existing estimators are not suitable for or give biased estimates for diffusion-based priors.

**Our contribution:** We propose the Diffusion Model Evidence (`DiME`) estimator, a method for estimating the model evidence under a diffusion model prior that does not require the prior score or density. We derive the estimator for posterior sampling methods that anneal along the *standard (posterior) marginals* $p(\boldsymbol{x}_t \mid \boldsymbol{y})$, as well as a generalized estimator for any arbitrary path of marginals that anneal to the true posterior. By integrating along these posterior time-marginals, `DiME` accurately estimates the model evidence using intermediate samples already generated during posterior sampling. `DiME` thus only requires a small number of posterior samples (e.g., 20) for accurate estimation. We also describe how to practically implement `DiME` alongside a state-of-the-art posterior sampling method Decoupled Annealing Posterior Sampling (DAPS).

We perform several experiments validating our method. We first test in a mixture of Gaussian setting where an analytic evidence can be derived, and find that `DiME` provides nearly unbiased estimates of the evidence and performs comparably to baselines sequential Monte Carlo, annealed importance sampling, and thermodynamic integration despite not using the prior score. We also test `DiME` on two non-convex inverse problems, Gaussian and Fourier phase retrieval. `DiME` consistently selects the correct prior from a set of 10 diffusion models trained on MNIST (Deng, 2012) digits given a single noisy measurement of one MNIST digit, while the baselines fail to do so.

Finally, we use our method to perform both model selection and validation on real M87* black hole observations from the Event Horizon Telescope (EHTC, 2019b;c). `DiME` indicates that a prior derived from synthetic black hole images generated using General-Relativistic Magnetohydrodynamics (GRMHD) (Mizuno, 2022) has higher likelihood compared to the Radiatively Inefficient Accretion Flow (RIAF) (Broderick et al., 2011) black hole prior, a prior trained on general space images (Alam et al., 2024), a prior trained on faces (Liu et al., 2015), and a prior trained on MNIST digit 0s. Furthermore, we diagnose the validity of the GRMHD prior by prior predictive checking on the model evidence. Our results suggest that the M87* observations are statistically in-distribution with respect to the GRMHD prior, while still leaving room in the model for refinement.

## 2 BACKGROUND

### 2.1 DIFFUSION MODELS

Given a clean image $\boldsymbol{x}_0 \sim p_0(\boldsymbol{x}_0)$, the diffusion forward process at time $t \in [0, T]$ is given by:

$$\boldsymbol{x}_t = a_t \boldsymbol{x}_0 + \sigma_t \boldsymbol{z}_t, \quad \boldsymbol{z}_t \sim \mathcal{N}(0, \boldsymbol{I}) \tag{1}$$

where $a_t$ is the signal scaling, $\sigma_t$ is noise scaling, and $a_t'$, $\sigma_t'$ are the time derivatives. We can sample from the clean image distribution by discretizing the reverse SDE:

$$d\boldsymbol{x}_t = \left[ \frac{a_t'}{a_t} \boldsymbol{x}_t \ - \ g_t^2 \nabla_{\boldsymbol{x}_t} \log p_t(\boldsymbol{x}_t) \right] dt \ + \ g_t \, d\boldsymbol{w}_t \tag{2}$$

where $g_t^2 = 2 \left( \sigma_t \sigma_t' - \frac{a_t'}{a_t} \sigma_t^2 \right)$, $\boldsymbol{w}_t$ is Brownian motion and

$\nabla_{\boldsymbol{x}_t} \log p_t(\boldsymbol{x}_t)$ is learned. We can obtain the posterior mean via Tweedie's formula (Efron, 2011):

$$\mathbb{E}[\boldsymbol{x}_0 \mid \boldsymbol{x}_t] = \frac{1}{\alpha_t} \Big( \boldsymbol{x}_t + \sigma_t^2 \, \nabla_{\boldsymbol{x}_t} \log p(\boldsymbol{x}_t) \Big). \tag{3}$$

## 2.2 DIFFUSION-BASED POSTERIOR SAMPLING METHODS

Most diffusion posterior sampling methods fall broadly under two categories: (1) explicitly approximating the intractable likelihood score $\nabla_{\boldsymbol{x}_t} \log p_t(\boldsymbol{y} \mid \boldsymbol{x}_t)$ as guidance during reverse diffusion, and (2) methods that anneal along a path of specified intermediate marginals, using techniques such as sequential Monte Carlo (Cardoso et al., 2023; Wu et al., 2023), gradient-based optimization (Moufad et al., 2024), or Langevin dynamics (Zhang et al., 2025; Wu et al., 2024; Zhu et al., 2023). The approximation errors of (1) compound over the sampling process, resulting in intermediate time-marginals with no tractable form and a biased posterior, while (2) corrects these errors by adhering to a specified marginal path, allowing for their time-marginals to anneal to the true posterior. As a result, in this work we focus on two different sampling methods of (2) that have demonstrated strong performance for solving physical inverse problems (Zheng et al., 2025).

Concretely, given measurements $\boldsymbol{y}$ from an inverse problem $\boldsymbol{y} = \boldsymbol{A}(\boldsymbol{x}) + \varepsilon$ we introduce two likelihood terms. The first, $p(\boldsymbol{y} \mid \boldsymbol{x}_t) = \int p(\boldsymbol{y} \mid \boldsymbol{x}_0)p(\boldsymbol{x}_0 \mid \boldsymbol{x}_t)d\boldsymbol{x}_0$ is the marginal likelihood of $\boldsymbol{y}$ conditioned on the current diffusion state $\boldsymbol{x}_t$. The second, $q(\boldsymbol{y} \mid \boldsymbol{x}_t) := p(\boldsymbol{y} \mid \boldsymbol{x}_0 = \boldsymbol{x}_t)$ is a time-independent data likelihood term that assumes $\boldsymbol{x}_t$ is the clean image.

The *Decoupled Annealing Posterior Sampling* (DAPS) method (Zhang et al., 2025) alternates between noising and sampling from the posterior. Given initial noise samples $\boldsymbol{x}_T \sim \mathcal{N}(0, \boldsymbol{I})$ and $N$-step annealing schedule $[t_1, \ldots, t_N]$, we sample from the clean image posterior using Langevin dynamics before re-adding noise according to the diffusion process:

$$\tilde{\boldsymbol{x}}_0 \sim p(\boldsymbol{x}_0 \mid \boldsymbol{x}_{t_k}, \boldsymbol{y}) \propto p(\boldsymbol{y} \mid \boldsymbol{x}_0)p(\boldsymbol{x}_0 \mid \boldsymbol{x}_{t_k}) \tag{4}$$

$$\boldsymbol{x}_{t_{k-1}} \sim p(\boldsymbol{x}_{t_{k-1}} \mid \tilde{\boldsymbol{x}}_0) \tag{5}$$

Zhang et al. (2025) proposes two approaches: *Gaussian approximation DAPS*, where $p(\boldsymbol{x}_0 \mid \boldsymbol{x}_t)$ is approximated via a Gaussian $\mathcal{N}(\mathbb{E}[\boldsymbol{x}_0 \mid \boldsymbol{x}_t], \boldsymbol{\Sigma}_{\boldsymbol{x}_0 \mid \boldsymbol{x}_t})$ and the covariance $\boldsymbol{\Sigma}_{\boldsymbol{x}_0 \mid \boldsymbol{x}_t}$ is heuristically chosen as $\sigma_t^2$, and *exact DAPS*, where the prior score is computed using the diffusion model at a tiny noise level $\nabla \log p(\boldsymbol{x}_0) \approx \nabla \log p(\boldsymbol{x}, \sigma_{min})$. The unconditional distribution of intermediate samples closely match the *standard marginals*: $\boldsymbol{x}_t \sim p(\boldsymbol{x}_t \mid \boldsymbol{y}) \propto p(\boldsymbol{y} \mid \boldsymbol{x}_t)p(\boldsymbol{x}_t)$ for all times $t$.

The *Plug-and-Play Diffusion Models* (PnP-DM) sampling method (Wu et al., 2024) takes inspiration from the Split Gibbs Sampler (Vono et al., 2019) and related diffusion-based samplers (Coeurdoux et al., 2024) by splitting each annealing iteration into a likelihood step and a prior step. In particular, given $N$-step annealing schedule $[t_1, \ldots, t_N]$ and clean image initialization $\hat{\boldsymbol{x}}_0$, we alternate between running Langevin dynamics to obtain a sample from the next time-marginal posterior and using the diffusion model to sample a clean image. We rewrite the updates for one timestep $t_k$ in terms of standard diffusion notation:

$$\boldsymbol{x}_{t_k} \sim q(\boldsymbol{x}_{t_k} \mid \boldsymbol{y}, \hat{\boldsymbol{x}}_0) \propto q(\boldsymbol{y} \mid \boldsymbol{x}_{t_k})p(\boldsymbol{x}_{t_k} \mid \hat{\boldsymbol{x}}_0) \tag{6}$$

$$\hat{\boldsymbol{x}}_0 \sim p(\boldsymbol{x}_0 \mid \boldsymbol{x}_{t_k}) \tag{7}$$

where $\boldsymbol{x}_{t_k}$ is sampled from a joint distribution of the noised $\hat{\boldsymbol{x}}_0$ and the time-independent likelihood $q$. We write $q(\boldsymbol{y} \mid \boldsymbol{x}_{t_k})$ to emphasize this proxy likelihood applies the measurement model directly to a noisy rather than clean image.

We refer to the resulting unconditional distributions of intermediate samples as *PnP-DM marginals*, denoted by $\boldsymbol{x}_t \sim q(\boldsymbol{x}_t \mid \boldsymbol{y}) \propto q(\boldsymbol{y} \mid \boldsymbol{x}_t)p(\boldsymbol{x}_t)$ for all $t$.

## 2.3 EXISTING MODEL EVIDENCE ESTIMATORS

The simplest estimator approximates the evidence by averaging $p(\boldsymbol{y} \mid \boldsymbol{x})$ over samples $\boldsymbol{x} \sim p(\boldsymbol{x})$, though this requires exponentially many samples in high dimensions. The standard harmonic mean estimator (Newton & Raftery, 1994) mitigates this issue but has infinite variance. Learned harmonic

mean estimators (McEwen et al., 2021; Polanska et al., 2023; Spurio Mancini et al., 2023) stabilize the variance by importance sampling with a surrogate neural density, under the assumption that the prior density is known. However, surrogate training requires thousands of posterior samples and importance sampling requires a similar amount of density evaluations even in medium-dimensional settings, making these approaches infeasible for diffusion-based priors.

Nested sampling (Skilling, 2006; Cai et al., 2022) re-parametrizes the model evidence into a one-dimensional integral over the prior volume, but still requires the score or density and is limited to log-convex likelihoods. Thermodynamic integration (TI) (Lartillot & Philippe, 2006), annealed importance sampling (AIS) (Neal, 2001) and sequential Monte Carlo (SMC) (Del Moral et al., 2006) compute the evidence by integrating over a path of distributions $p_\beta$, with $0 \leq \beta \leq 1$. For diffusion priors, accurately sampling from arbitrary $p_\beta(\boldsymbol{x})$ is computationally difficult as we lack a clean prior score. Reverse diffusion Monte Carlo (RDMC) (Huang et al., 2023) methods simulate the diffusion process to sample from unnormalized densities, using a Monte Carlo approach to approximate the noised scores. Guo et al. (2025) independently derived an evidence estimator for the reverse diffusion process under very different assumptions: they rely on a closed-form posterior score, obtain exact posterior marginals directly from an analytic diffusion that is not learned, and provide only theoretical convergence results without experimental validation.

## 3 ESTIMATING THE MODEL EVIDENCE ALONG THE STANDARD MARGINALS

We introduce `DiME` to estimate model evidence from the standard marginals $p(\boldsymbol{x}_t \mid \boldsymbol{y})$, aligning closely with the annealing path of DAPS. All proofs in this section are shown in Appendix A. A general model evidence estimator for arbitrary marginals annealing to the true posterior is derived in Appendix B, and an unnormalized model evidence estimator that instead uses the PnP-DM marginals (`DiME-PnPDM`) is derived and implemented in Appendix B.1.

**Proposition 1** (`DiME`: Model evidence estimation along the standard marginals). *Given diffusion process $\boldsymbol{x}_t = a_t \boldsymbol{x}_0 + \sigma_t \boldsymbol{z}_t, \boldsymbol{z}_t \sim \mathcal{N}(0, \boldsymbol{I})$, posterior marginals $p(\boldsymbol{x}_t \mid \boldsymbol{y}) \propto p(\boldsymbol{x}_t)p(\boldsymbol{y} \mid \boldsymbol{x}_t)$, and timesteps $0 = t_0 < \cdots < t_N = T$, the model evidence can be estimated via:*

$$\log p(\boldsymbol{y}) = \mathbb{E}_{\boldsymbol{x}_0 \sim p(\boldsymbol{x}_0 \mid \boldsymbol{y})}\big[ \log p(\boldsymbol{y} \mid \boldsymbol{x}_0) \big] - D_{\mathrm{KL}}(p(\boldsymbol{x}_0 \mid \boldsymbol{y}) || p(\boldsymbol{x}_0)). \tag{8}$$

*The log-likelihood term can be estimated with posterior samples [1], and the KL divergence with:*

$$D_{\mathrm{KL}}(p(\boldsymbol{x}_0 \mid \boldsymbol{y}) || p(\boldsymbol{x}_0)) \approx \sum_{i=1}^{N} c_{t_i} \Delta t_i \mathbb{E}_{\boldsymbol{x}_{t_i} \sim p(\boldsymbol{x}_{t_i} \mid \boldsymbol{y})} \| \nabla_{\boldsymbol{x}_{t_i}} \log p(\boldsymbol{y} \mid \boldsymbol{x}_{t_i}) \|^2 \tag{9}$$

*where $c_{t_i} = \sigma'_{t_i}\sigma_{t_i} - \sigma_{t_i}^2 \frac{a'_{t_i}}{a_{t_i}}$ and $\Delta t_i = t_i - t_{i-1}$ are derived from the diffusion schedule.*

The sum across each sample-path $\{\boldsymbol{x}_{t_i}\}_{i=0,\ldots,N}$ can be viewed as the distance of the resulting posterior sample $\boldsymbol{x}_0$ to the prior. In the remainder of the section, we showcase how to practically implement `DiME` in tandem with the DAPS method but with two modifications: in Section 3.1 we design an improved covariance approximation[2] for $p(\boldsymbol{x}_0 \mid \boldsymbol{x}_t)$ if the Gaussian approximation DAPS is used, and in Section 3.2 we propose to sample two $\tilde{\boldsymbol{x}}_0 \sim p(\boldsymbol{x}_0 \mid \boldsymbol{x}_t, \boldsymbol{y})$ for each $\boldsymbol{x}_t$ to estimate $\| \nabla_{\boldsymbol{x}_t} \log p(\boldsymbol{y} \mid \boldsymbol{x}_t) \|^2$ needed for Eq. 9. The full algorithm can be seen in Algorithm 1.

### 3.1 AN IMPROVED APPROXIMATION FOR THE POSTERIOR COVARIANCE

While the exact DAPS sampler can be used for accurate sampling of $\tilde{\boldsymbol{x}}_0 \sim p(\boldsymbol{x}_0 \mid \boldsymbol{x}_t, \boldsymbol{y}) \propto p(\boldsymbol{y} \mid \boldsymbol{x}_0)p(\boldsymbol{x}_0 \mid \boldsymbol{x}_t)$ (Eq. 4), Gaussian approximation DAPS provides a significantly cheaper alternative by approximating $p(\boldsymbol{x}_0 \mid \boldsymbol{x}_t) \approx \mathcal{N}(\mathbb{E}[\boldsymbol{x}_0 \mid \boldsymbol{x}_t], \sigma_t^2)$. However, this covariance heuristic only accounts for $p(\boldsymbol{x}_t \mid \boldsymbol{x}_0)$ and ignores $p(\boldsymbol{x}_0)$. While this heuristic is accurate at low noise levels, it overestimates the variance in the high noise regimes: at the terminal noise level $t = T$, we expect $p(\boldsymbol{x}_0 \mid \boldsymbol{x}_T) \approx p(\boldsymbol{x}_0)$ with a covariance that should match the prior $\mathrm{Cov}(\boldsymbol{x}_0)$, but the heuristic gives

---

[1] For likelihoods with Gaussian noise, if the posterior and prior have enough overlap such that posterior samples have a data fit of $\chi^2 \approx 1$, this term essentially becomes constant under the same likelihood function.

[2] While approximating $p(\boldsymbol{x}_0 \mid \boldsymbol{x}_t)$ is necessary for the standard marginals annealing path, it is not necessary for many annealing paths under the generalized framework introduced in Appendix B.

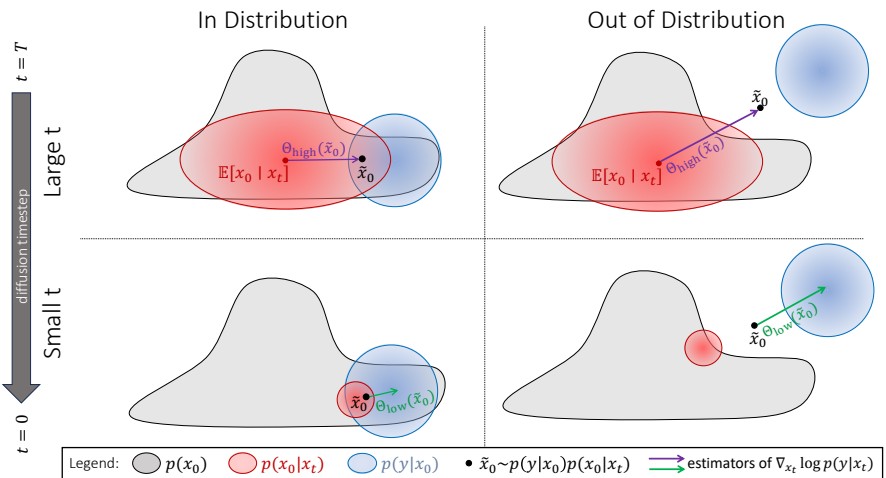

Figure 1: Visualization of the unbiased estimators $\Theta_{high}, \Theta_{low}$ of the likelihood score $\nabla_{\boldsymbol{x}_t} \log p(\boldsymbol{y} \mid \boldsymbol{x}_t)$ described in Lemma 2 for both in-distribution (left) and out-of-distribution (right) measurements $\boldsymbol{y}$. At large diffusion time steps, the high noise estimator (top) uses the distance between $\tilde{\boldsymbol{x}}_0$ and $\mathbb{E}[\boldsymbol{x}_0 \mid \boldsymbol{x}_t]$. At small diffusion time steps, the low noise estimator (bottom) uses the likelihood score at $\tilde{\boldsymbol{x}}_0$. The estimator is larger (longer arrows) when $\boldsymbol{y}$ is out-of-distribution, as there greater KL divergence from the posterior to the prior; this results in DiME computing a lower evidence. Note the scaling factors $\frac{a_t}{\sigma_t^2}$ and $\boldsymbol{\Sigma}_{\boldsymbol{x}_0 \mid \boldsymbol{x}_t}$ of the estimator are not shown.

$\text{Cov}(\boldsymbol{x}_0 \mid \boldsymbol{x}_T) \approx \sigma_T^2 \gg \text{Cov}(\boldsymbol{x}_0)$. This mismatch at high noise was also observed empirically in Zhang et al. (2025). Since DiME uses all time-marginals it is important to accurately estimate $p(\boldsymbol{x}_0 \mid \boldsymbol{x}_t)$ for all $t$. Therefore, we include knowledge of $p(\boldsymbol{x}_0)$ by approximating it as Gaussian $\mathcal{N}(\boldsymbol{\mu}_0, \boldsymbol{\Sigma}_0)$, using an empirical covariance computed from the training data similar to the covariance approximation introduced in Linhart et al. (2024). Introducing an approximate prior provides a more accurate representation of $p(\boldsymbol{x}_0 \mid \boldsymbol{x}_t)$, which remains a Gaussian approximation as it is now the product of two Gaussians. The resulting covariance is given by:

**Lemma 1.** *Suppose we have diffusion process $\boldsymbol{x}_t = a_t \boldsymbol{x}_0 + \sigma_t \boldsymbol{z}_t, \boldsymbol{z}_t \sim \mathcal{N}(0, \boldsymbol{I})$. Let $\boldsymbol{x}_t \sim p(\boldsymbol{x}_t)$ and suppose $p(\boldsymbol{x}_0) \approx \mathcal{N}(\boldsymbol{\mu}_0, \boldsymbol{\Sigma}_0)$. Then we can express the posterior covariance $\boldsymbol{\Sigma}_{\boldsymbol{x}_0 \mid \boldsymbol{x}_t}$ as*

$$\boldsymbol{\Sigma}_{\boldsymbol{x}_0 \mid \boldsymbol{x}_t} = \left[ \boldsymbol{\Sigma}_0^{-1} + \frac{a_t^2}{\sigma_t^2} \boldsymbol{I} \right]^{-1}. \tag{10}$$

For large images where computing a full covariance matrix is intractable, a diagonal approximation in a transform basis has also been demonstrated to work well (Peng et al., 2024), and for linear inverse problems, an analytic local covariance is computable (Boys et al., 2023). Our approximation aligns the annealing path more closely with $p(\boldsymbol{x}_t \mid \boldsymbol{y})$ for all $t$. It is worth noting that DAPS re-samples at every annealing step, ensuring that these Gaussian approximation errors are never compounded. As shown in Sections 4.1 and 4.3.1, using the proposed improved approximation leads to nearly unbiased evidence estimates for both a multimodal Gaussian toy example with varying local curvature as well as real-world scientific priors.

### 3.2 ESTIMATING THE SQUARED LIKELIHOOD SCORE

While directly computing $\nabla_{\boldsymbol{x}_t} \log p(\boldsymbol{y} \mid \boldsymbol{x}_t)$ is intractable, DiME only requires an unbiased estimate as seen in Eq. 9. Estimators using unconditional samples of $\boldsymbol{x}_0 \mid \boldsymbol{x}_t$ are high-variance: these samples usually have low likelihood $p(\boldsymbol{y} \mid \boldsymbol{x}_0)$ resulting in volatile scores. Thankfully, DAPS provides samples of $\tilde{\boldsymbol{x}}_0 \sim p(\boldsymbol{x}_0 \mid \boldsymbol{x}_t, \boldsymbol{y})$, which we can use to create estimators with lower variance:

**Lemma 2.** *Suppose we have diffusion process $\boldsymbol{x}_t = a_t \boldsymbol{x}_0 + \sigma_t \boldsymbol{z}_t, \boldsymbol{z}_t \sim \mathcal{N}(0, \boldsymbol{I})$ and sample $\tilde{\boldsymbol{x}}_0 \sim p(\boldsymbol{x}_0 \mid \boldsymbol{x}_t, \boldsymbol{y})$. Under the assumption that $p(\boldsymbol{x}_0 \mid \boldsymbol{x}_t) \approx \mathcal{N}(\mathbb{E}[\boldsymbol{x}_0 \mid \boldsymbol{x}_t], \boldsymbol{\Sigma}_{\boldsymbol{x}_0 \mid \boldsymbol{x}_t})$, both $\Theta_{high}(\tilde{\boldsymbol{x}}_0)$*

and $\Theta_{low}(\tilde{\boldsymbol{x}}_0)$ are unbiased estimators of $\nabla_{\boldsymbol{x}_t} \log p(\boldsymbol{y} \mid \boldsymbol{x}_t)$ with

$$\Theta_{high}(\tilde{\boldsymbol{x}}_0) = \frac{a_t}{\sigma_t^2} \left( \tilde{\boldsymbol{x}}_0 - \mathbb{E}[\boldsymbol{x}_0 \mid \boldsymbol{x}_t] \right). \tag{11}$$

$$\Theta_{low}(\tilde{\boldsymbol{x}}_0) = \frac{a_t}{\sigma_t^2} \left( \boldsymbol{\Sigma}_{\boldsymbol{x}_0 \mid \boldsymbol{x}_t} \nabla_{\tilde{\boldsymbol{x}}_0} \log p(\boldsymbol{y} \mid \tilde{\boldsymbol{x}}_0) \right). \tag{12}$$

*Under low diffusion noise* $(a_t \to 1, \sigma_t \to 0)$, $\mathrm{Var}(\Theta_{low}) \to 0$ *but* $\mathrm{Var}(\Theta_{high}) = O(\frac{1}{\sigma_t^2})$.

*Under high noise* $(a_t \to 0, \sigma_t \to 1)$, $\mathrm{Var}(\Theta_{high}) \to 0$ *but* $\mathrm{Var}(\Theta_{low}) = O(\frac{\sigma_t^4}{\sigma_{\boldsymbol{y}}^4} \mathrm{Var}(\Theta_{high}))$.

A proof of these bounds is displayed in Appendix A.1. The lack of the measurement $\boldsymbol{y}$ in the high noise estimator bounds the worst possible case, where the data is weak; when the measurement contains a lot of information, the variance would be much lower. As such, $\Theta_{high}$ is used more towards the beginning of the sampling process while $\Theta_{low}$ is used towards the end. As both estimators are cheap to compute, we can simply evaluate both and choose the lower variance estimator at each timestep. Finally, as we want an unbiased estimate of the *squared* likelihood score, squaring the estimator is biased: $\mathbb{E}\|\Theta\|^2 = \|\mathbb{E}\Theta\|^2 + \mathrm{Tr}(\mathrm{Cov}(\Theta))$. Therefore, for each intermediate sample $\boldsymbol{x}_t$, we propose to sample two i.i.d $\tilde{\boldsymbol{x}}_0^{(1)}, \tilde{\boldsymbol{x}}_0^{(2)} \sim p(\boldsymbol{x}_0 \mid \boldsymbol{x}_t, \boldsymbol{y})$, giving us the following unbiased estimator: $\Theta(\tilde{\boldsymbol{x}}_0^{(1)})^T \Theta(\tilde{\boldsymbol{x}}_0^{(2)})$. The full method is displayed in Algorithm 1, and a visualization for both in-distribution and out-of-distribution measurements is visualized in Figure 1.

---

**Algorithm 1** One posterior sample-path of `DiME` implemented with DAPS

---

**Require:** Score-based model $s_\theta$, measurement $\boldsymbol{y}$, noise schedule $\sigma_t$, $(t_i)_{i \in \{1, \dots, N\}}$
 1: Sample $\boldsymbol{x}_{t_N} \sim \mathcal{N}(0, \sigma_{t_N}^2 \boldsymbol{I})$.
 2: Initialize the integral sum, $f = 0$.
 3: **for** $i = N, N-1, \dots, 1$ **do**
 4:     Compute $\hat{\boldsymbol{x}} = \mathbb{E}[\boldsymbol{x}_0 \mid \boldsymbol{x}_t]$ using Tweedie's formula and $s_\theta$.
 5:     Sample $\tilde{\boldsymbol{x}}_0^{(1)}, \tilde{\boldsymbol{x}}_0^{(2)} \sim p(\boldsymbol{x}_0 \mid \boldsymbol{x}_{t_i}, \boldsymbol{y})$ using Langevin with $p(\boldsymbol{x}_0 \mid \boldsymbol{x}_t) = \mathcal{N}(\hat{\boldsymbol{x}}, \boldsymbol{\Sigma}_{\boldsymbol{x}_0 \mid \boldsymbol{x}_t})$.   (3.1)
 6:     Sample $\boldsymbol{x}_{t_{i-1}} \sim \mathcal{N}(a_{t_{i-1}} \tilde{\boldsymbol{x}}_0^{(1)}, \sigma_{t_{i-1}}^2 I)$.
 7:     $f_i = \arg\min_{\Theta \in \{\Theta_{\text{low}}, \Theta_{\text{high}}\}} \mathrm{Var}(\Theta(\tilde{\boldsymbol{x}}_0^{(1)})^T \Theta(\tilde{\boldsymbol{x}}_0^{(2)}))$ computed using all sample-paths.   (3.2)
 8:     Update the integral sum $f \leftarrow f + (t_i - t_{i-1}) \left( \sigma_t' \sigma_t - \sigma_t^2 \frac{a_t'}{a_t} \right) f_i$.
 9: **end for**
10: **return** $\boldsymbol{x}_0, -f + \log p(\boldsymbol{y} \mid \boldsymbol{x}_0)$

---

## 4 EXPERIMENTS

We compare `DiME` and `DiME-PnPDM` with five baselines: naive Monte Carlo (Naive MC), thermodynamic integration (TI), annealed importance sampling (AIS), Sequential Monte Carlo (SMC), and the original DAPS covariance heuristic (Original DAPS heuristic). Baseline methods are described in Appendix C and all tuning and implementation details are provided in Appendix D.

### 4.1 BENCHMARKING ON GAUSSIAN MIXTURE PRIOR WITH GROUND TRUTH EVIDENCE

To verify `DiME`, we first test on a multimodal 1000-D Gaussian mixture prior and linear forward model $\boldsymbol{y} = \boldsymbol{A}\boldsymbol{x} + \boldsymbol{\varepsilon}, \boldsymbol{\varepsilon} \sim \mathcal{N}(0, \sigma^2 \boldsymbol{I})$ where the model evidence has an analytic form to compare to. We use two Gaussian components with means at $(-\mathbf{0.75}, \mathbf{0.75})$ and identical covariances of $0.25\boldsymbol{I}$. The entries of $\boldsymbol{A} \in \mathbb{R}^{200 \times 1000}$ are i.i.d. $\mathcal{N}(0, \frac{1}{200})$, and $\sigma = 0.1$. We run three experiments where the ground truth $\boldsymbol{x}^*$ has varying model evidence to test the robustness of our method: in-distribution $\boldsymbol{x}^*$, out-of-distribution $\boldsymbol{x}^*$, and $\boldsymbol{x}^* = 0$ located at a saddle point. The analytic $\nabla_{\boldsymbol{x}} \log p(\boldsymbol{x})$ is used to sample the intermediate distributions for TI, AIS, and SMC. In contrast, for `DiME` the analytic $\nabla_{\boldsymbol{x}_t} \log p(\boldsymbol{x}_t)$ is used to compute $\mathbb{E}[\boldsymbol{x}_0 \mid \boldsymbol{x}_t]$ for $\sigma_t > 0.05$ and the analytic score is never used. All methods are tested using 100 discretization steps (equivalently, annealing time steps for `DiME`) and 20 sample paths for 50 trials. The mean and standard deviation is reported.

| | In-distribution $x^*$ | | Out-of-distribution $x^*$ | | Saddle point $x^*$ | |
|---|---|---|---|---|---|---|
| | Estimate | Rel. Error ↓ | Estimate | Rel. Error ↓ | Estimate | Rel. Error ↓ |
| **Ground truth** $\log p(\boldsymbol{y})$ | $-128.7 \pm 0.0$ | 0.0 | $-1680.2 \pm 0$ | 0.0 | $-251.3 \pm 0.0$ | 0.0 |
| Naive MC (1000 samples) | $-3283 \pm 130$ | 2451% | $-41289 \pm 577$ | 2357% | $-6029 \pm 170$ | 2299% |
| Naive MC (10000 samples) | $-3041 \pm 113$ | 2263% | $-40123 \pm 487$ | 2288% | $-5666 \pm 144$ | 2155% |
| Original DAPS heuristic | $-316.2 \pm 79.5$ | 146% | $-1735.4 \pm 4.7$ | 3.3% | $-269.6 \pm 3.3$ | 7.3% |
| TI | $-132.8 \pm 1.3$ | 3.2% | $-1773.8 \pm 11.8$ | 5.6% | $-254.2 \pm 3.2$ | 1.2% |
| AIS | $-135.9 \pm 2.5$ | 5.6% | $-1754.6 \pm 7.1$ | 4.4% | $-259.1 \pm 2.9$ | 3.1% |
| SMC | $-132.1 \pm 1.0$ | 2.6% | $-1701.1 \pm 5.8$ | 1.2% | $-253.0 \pm 3.4$ | **0.7**% |
| DiME | $-126.8 \pm 2.8$ | **1.5**% | $-1674.8 \pm 4.7$ | **0.3**% | $-253.3 \pm 3.1$ | 0.8% |
| DiME-PnPDM * | $-131.8 \pm 2.0$ | 2.4% | $-1912.1 \pm 6.5$ | 13.8% | $-245.9 \pm 3.6$ | 2.1% |

Table 1: Evidence estimates on Mixture of Gaussians study with analytic $\log p(\boldsymbol{y})$ and ground truth $x^*$ in-distribution (top), out-of-distribution (middle), and at a saddle point (bottom). Mean and standard deviation of 50 trials displayed each with 20 sample paths. When the analytic prior score is available, DiME performs comparably to the baselines for all cases, while DiME-PnPDM performs well when the ground truth is roughly in-distribution. **\*Important:** as DiME-PnPDM outputs an unnormalized estimate, we normalize via the analytic constant factor for ease of comparison.

Results are displayed in Table 1. DiME gives nearly unbiased estimates of the evidence in all cases, with comparable performance to the strongest baseline (SMC), despite never using the true prior score. To probe the effect of the covariance choice discussed in Sec. 3.1, we first consider the case where $x^*$ lies within a prior mode (in-distribution). Here the original DAPS heuristic consistently overestimates the variance of the marginals, pushing posterior samples into the wrong mode and inducing a large bias in the evidence estimate. Even when $x^*$ is outside any single mode, the original heuristic continues to overestimate the evidence. In contrast, our improved covariance heuristic, incorporated in DiME, always generates samples from the correct mode and eliminates this bias, even when the true $x^*$ lies in regions of local curvature far from the unimodal Gaussian assumption.

Unlike DiME, DiME-PnPDM only performs well for roughly in-distribution tasks. For out-of-distribution measurements $\boldsymbol{y}$, while the PnP-DM marginals asymptotically anneal to the correct posterior, we find that in the finite-step case this annealing path results in greater bias than DAPS. Because out-of-distribution measurements frequently arise in model selection, we conclude that PnP-DM is less well suited for this task.

## 4.2 Model Selection Using Learned Score on Non-convex Inverse Problems

We test both DiME and the SMC baseline on two non-convex inverse problems, Gaussian phase retrieval and Fourier phase retrieval. In Gaussian phase retrieval, the forward model is $\boldsymbol{y} = |\boldsymbol{Ax}| + \varepsilon$, where $\boldsymbol{A} \in \mathbb{C}^{m \times n}$ consists of i.i.d Gaussian $\mathcal{N}(0, \frac{1}{m})$ and $\varepsilon \sim \mathcal{N}(0, \sigma^2 \boldsymbol{I})$. In Fourier phase retrieval, the forward model is $\boldsymbol{y} = |\boldsymbol{Fx}| + \varepsilon, \varepsilon \sim \mathcal{N}(0, \sigma^2 \boldsymbol{I})$ where $\boldsymbol{F}$ is the discrete Fourier transform. In our experiments, we use $m = 0.1n$ and $\sigma = 0.3$. While Fourier phase retrieval has many more measurements, the structured forward operator means that both translation and flips of $\boldsymbol{x}$ are invariances under the forward model. We test with ten diffusion model priors, one on each MNIST (Deng, 2012) digit, and compute the evidence for each prior given a measurement of each digit. For DiME, we use 50 annealing iterations and 20 generated sample-paths for each entry. For SMC, we use the learned $\nabla_{\boldsymbol{x}} \log p(\boldsymbol{x}_t)$ as a surrogate prior score at different low noise levels.

DiME results are shown in Figure 2, and SMC results are displayed in Appendix E. For both inverse problems, DiME always selects the correct model given a single noisy measurement $\boldsymbol{y}$, while SMC often fails to do so, demonstrating that methods relying on a clean prior score are not suitable for diffusion-based model selection. For Gaussian phase retrieval, DiME predicts higher likelihoods of digits that look similar, such as 4 and 9 (Figure 2, bottom left, third row). For Fourier phase retrieval, DiME predicts high evidence for the MNIST 9 prior when the ground truth image is a 6 (and vice versa), but as both vertical and horizontal flips are invariances under Fourier phase retrieval, performing these operations on a 6 results in a 9 (Figure 2, bottom right, second row). The normalized evidence can also provide useful insights: as the MNIST digit 1 prior has higher evidence for in-distribution $\boldsymbol{y}$ and lower evidence for out-of-distribution $\boldsymbol{y}$, it must be a narrower prior. On the other hand, the MNIST 3, 5 and 8 priors have higher out-of-distribution model evidence, suggesting that they are wider priors, likely due to the variation in which humans write these digits.

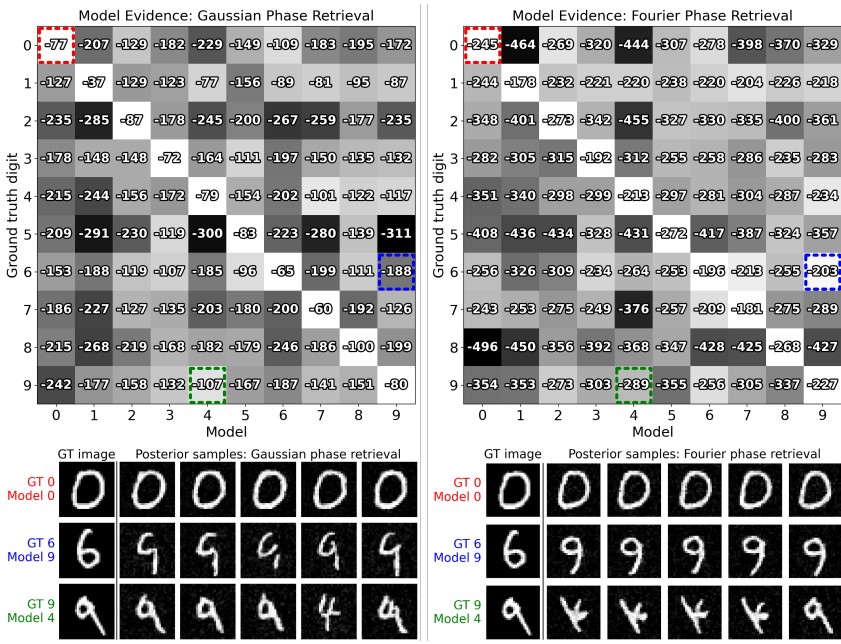

Figure 2: Model evidence confusion matrix for Gaussian phase retrieval (**left**) and Fourier phase retrieval (**right**) for each (ground truth measurement, model) pair of MNIST digits. Our method selects the correct model for all cases. Posterior samples shown for dotted matrix entries below. For Gaussian phase retrieval, DiME estimates higher model likelihood for visually similar digits, such as 4 and 9. For Fourier phase retrieval, both translations and reflections are invariances as seen in the posterior samples, so DiME estimates a high likelihood of model 9 given a measurement of a 6.

### 4.3 EVALUATING PRIOR MODEL EVIDENCE FOR REAL M87* BLACK HOLE DATA

In this section, we perform model selection and validation on real black hole M87* observations from April 6, 2017 (EHTC, 2019a;b;c). The need for such analysis was first emphasized during the release of the initial M87* images by the Event Horizon Telescope Collaboration (EHTC, 2019e), where the goal was not only to reconstruct images but also to test which simple parametric models best matched the data—particularly whether a ring provided the most suitable description (EHTC, 2019e; Broderick et al., 2020). In addition, the team directly compared the M87* data to a discrete set of synthetic black hole simulations (GRMHD) to assess which physical model parameters were most consistent with the observations (Fromm et al., 2019; EHTC, 2019e;d). However, as noted in EHTC (2019e), these conclusions are only valid to the extent that the selected simulations provide an adequate description of M87* – a question that can be answered by our proposed method.

Measurements, known as *visibilities*, are obtained by combining signals collected by telescopes observing simultaneously at different locations around the world. The forward model for complex visibilities between two telescopes $i$, $j$ is $\boldsymbol{v}_{i,j} = g_i g_j e^{i(\phi_i - \phi_j)} \boldsymbol{v}_{i,j}^* + \boldsymbol{\varepsilon}_{i,j}$, where $\boldsymbol{v}_{i,j}^*$ is the true visibilities measurements, $g_i$ is a station-dependent gain error, $\phi_i$ is a station-dependent phase error arising from atmospheric noise and $\boldsymbol{\varepsilon}_{i,j}$ is Gaussian thermal noise. To become invariant to the phase error, we use the closure phase $\boldsymbol{y}_{cp} = \boldsymbol{v}_{i,j} \boldsymbol{v}_{j,k} \boldsymbol{v}_{k,i}$ for a minimal set of 3 telescopes $(i, j, k)$ and for similar invariance to the gain error, we use the closure amplitude $\boldsymbol{y}_{ca} = \frac{\boldsymbol{v}_{i,j} \boldsymbol{v}_{k,l}}{\boldsymbol{v}_{i,k} \boldsymbol{v}_{j,l}}$ for a minimal set of 4 telescopes $(i, j, k, l)$ (Thompson et al., 2017; Blackburn et al., 2020). We use the InverseBench (Zheng et al., 2025) library to compute the likelihood. We generate 20 posterior samples using 50 annealing steps and 1000 Langevin dynamics steps per iteration. Gaussian approximation DAPS takes around 4 minutes to compute, while exact DAPS takes around 30 minutes to compute for the same parameters, a speedup of around 7x.

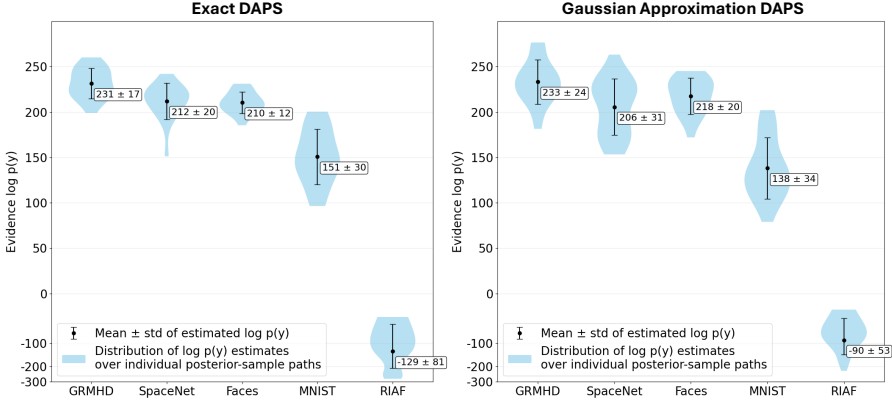

Figure 3: Model evidence estimates on real M87* observations across 5 different priors using exact DAPS (**left**) and Gaussian approximation DAPS (**right**). Our method concludes that, of these prior models, GRMHD is the most likely model. The violin plots show the distribution of evidences from 20 different posterior-sample paths from the DAPS sampling process, and the mean is the overall evidence estimate. Gaussian approximation DAPS gives highly accurate evidence estimates with 7x less compute than exact DAPS, but with slightly higher variance.

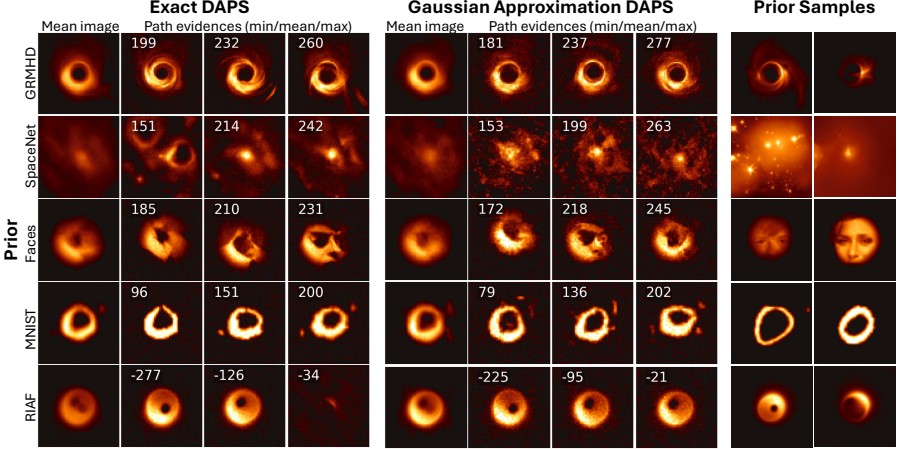

Figure 4: M87* reconstructions using the 5 priors using exact DAPS (**left**) and Gaussian approximation DAPS (**middle**). While all posterior samples have a data fit of around reduced $\chi^2 \approx 1$, they have varying path evidence estimates (shown in the top left of each image). **Right:** Example unconditional samples from each prior.

### 4.3.1 MODEL SELECTION

We trained five candidate diffusion priors: (1) on 48000 General-Relativistic Magnetohydrodynamic (GRMHD) simulations (Mizuno, 2022), (2) on 9070 Radiatively Inefficient Accretion Flow (RIAF) simulations (Broderick et al., 2011), (3) on 11000 general space images (SpaceNet) (Alam et al., 2024), (4) on 5923 MNIST (Deng, 2012) digit 0 images, and (5) on 48000 radially tapered CelebA Liu et al. (2015) images (Faces). Additionally, we augment our data during training to improve generality and robustness of our priors. The GRMHD images are randomly flipped horizontally and/or vertically and zoomed in and out in the range $[0.75, 1.25]$. The RIAF images were randomly flipped, zoomed in and out in the range $[0.75, 1.25]$, and rotated uniformly in the range $[0, 2\pi]$. The MNIST digit 0 images were zoomed in and out in the range $[0.5, 1]$.

Model selection results for the real M87* data can be seen in Figure 3. The mean reconstruction and posterior samples with the minimum, mean and maximum path evidence are shown on in Figure 4. Synthetic tests validating the method on this problem are shown in Appendix F. `DiME` determines that, for static imaging, GRMHD is the most likely prior for the M87* observations, followed by

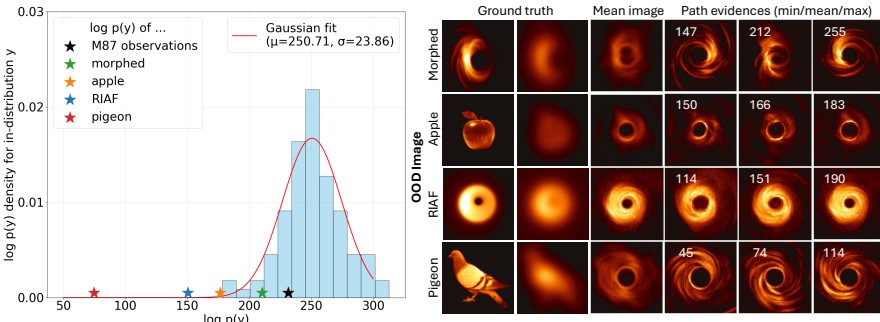

Figure 5: **Left:** GRMHD model validation results on M87* observations by comparing to evidence of in-distribution measurements $y$. Our method shows that the evidence of M87* observations have a $z$-score of about -0.81 compared to the evidence distribution of GRMHD measurements, indicating that M87* is statistically in-distribution of GRMHD. The evidence of simulated measurements of out-of-distribution images are also shown, demonstrating that measurements from out-of-distribution (OOD) images have OOD evidence. **Right:** Mean reconstruction and posterior samples of the OOD images with the highest and lowest distance to the GRMHD prior. The blurred ground truth image, or the maximum resolution that can be obtained from measurements, is also displayed.

SpaceNet, Faces, MNIST, and finally RIAF. While SpaceNet contains a few black hole images in its training data, our method penalizes it for being too general, as demonstrated with its wide range of unconditional samples (Figure 3, right, second row). The RIAF model is a very narrow prior that enforces a strong assumption on the structure of black holes, so any out-of-distribution images (such as M87* as determined by our method) will have low likelihood. We also find that Gaussian approximation DAPS gives nearly identical estimates as exact DAPS with 7x less compute.

### 4.3.2 MODEL VALIDATION

We also diagnose the validity of the GRMHD model for static imaging of the M87* observations by computing the evidence $\log p(y)$. We simulated measurements from 100 GRMHD test images using the same forward model as the EHT observations of M87*, and then computed the evidence for these measurements. Under a central limit theorem argument, we can approximate this distribution as Gaussian. We then compared the evidence of the true M87* observation to this Gaussian as well as the evidence of out-of-distribution measurements under the same forward model to assess if the GRMHD prior is valid for M87*.

Results are shown in Figure 5. The fitted Gaussian has $\mu = 250.71$ and $\sigma = 23.86$. M87* has an evidence of 231.46, giving it a z-score of -0.81 and p-value of 0.209 with respect to the previous Gaussian fit. Therefore, M87* likely lies within the support of the GRMHD image prior, offering credence to our current physical model of black holes through the images it produces, while still leaving open the possibility of an improved model. On the other hand, the morphed GRMHD image has evidence 210.37, giving it a z-score of -1.69 and p-value of 0.046, suggesting that it is not in the true GRMHD distribution. The apple, RIAF, and pigeon images have even lower evidences of 176.52, 150.41, and 74.68 respectively and are all considered out-of-distribution.

## 5 CONCLUSION

We introduce `DiME`, a method that estimates the Bayesian model evidence when solving inverse problems using diffusion model priors. `DiME` computes the KL divergence between the posterior and prior by integrating along posterior time-marginals, a task made easy by using the intermediate samples of recent diffusion posterior sampling methods. Empirically, `DiME` outperforms all baseline methods and we demonstrate how `DiME` can be used on a real-world black hole imaging problem. Altogether, `DiME` makes it possible to harness diffusion priors not only for reconstruction but also for principled model selection and validation, laying the foundation for more reliable inference in scientific imaging.

ETHICS STATEMENT

Our method is a statistical estimator and can occasionally select the incorrect model, potentially leading to incorrect conclusions. Furthermore, while it could in principle be applied to sensitive characteristics (e.g., facial recognition), we do not support its use in this context.

REPRODUCIBILITY STATEMENT

All code and trained models are available at https://github.com/fredwang25/DiME. All mathematical derivations are detailed in the appendix. The MNIST dataset is available through torchvision (Marcel & Rodriguez, 2010), the CelebA dataset is available at https://mmlab.ie.cuhk.edu.hk/projects/CelebA.html, and the SpaceNet dataset is available at https://www.kaggle.com/datasets/razaimam45/spacenet-an-optimally-distributed-astronomy-data/. The RIAF and GRMHD datasets are not currently available for public use.

ACKNOWLEDGMENTS

This work was supported by NSF Award 2048237, an Amazon AI4Science Discovery Award, OpenAI, and a Sloan Research Fellowship. FW is supported by a Kortschak Fellowship. The authors would like to thank Angela Gao for helpful discussions.

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

## A    PROOFS

Although the main text uses the diffusion SDE formulation, we introduce the Probability Flow ODE (PF-ODE) for the proofs and for Appendix B. This ODE has identical marginals as the SDE under ideal conditions. Given diffusion process $\boldsymbol{x}_t = a_t\boldsymbol{x}_0 + \sigma_t\boldsymbol{z}_t, \boldsymbol{z}_t \sim \mathcal{N}(0, \boldsymbol{I})$, the PF-ODE is:

$$d\boldsymbol{x}_t \;=\; \left[\frac{a'_t}{a_t}\,\boldsymbol{x}_t \;-\; \left(\sigma_t\sigma'_t - \frac{a'_t}{a_t}\sigma_t^2\right)\nabla_{\boldsymbol{x}_t}\log p(\boldsymbol{x}_t)\right]dt = \boldsymbol{v}_p(\boldsymbol{x}_t)\,dt \tag{13}$$

where $\boldsymbol{v}_p(\boldsymbol{x}_t)$ is the unconditional flow. The conditional flow $\boldsymbol{v}_p(\boldsymbol{x}_t \mid \boldsymbol{y})$ can be derived via Bayes:

$$\boldsymbol{v}_p(\boldsymbol{x}_t \mid \boldsymbol{y}) = \frac{a'_t}{a_t}\,\boldsymbol{x}_t \;-\; \left(\sigma_t\sigma'_t - \frac{a'_t}{a_t}\sigma_t^2\right)\nabla_{\boldsymbol{x}_t}\log p(\boldsymbol{x}_t \mid \boldsymbol{y}) \tag{14}$$

$$= \boldsymbol{v}_p(\boldsymbol{x}_t) - \left(\sigma'_t\sigma_t - \sigma_t^2\frac{a'_t}{a_t}\right)\nabla_{\boldsymbol{x}_t}\log p(\boldsymbol{y} \mid \boldsymbol{x}_t). \tag{15}$$

For the following proof, we impose the regularity and smoothness assumptions listed in Appendix A of Song et al. (2021a) for both the prior marginals $p(\boldsymbol{x}_t)$ and the posterior marginals $\pi(\boldsymbol{x}_t \mid \boldsymbol{y})$.

**Lemma 3.** *Let $p(\boldsymbol{x}_t)$ be the unconditional diffusion marginal and $\pi(\boldsymbol{x}_t \mid \boldsymbol{y}) \propto p(\boldsymbol{x}_t)\pi(\boldsymbol{y} \mid \boldsymbol{x}_t)$ be a posterior time-marginal obtained at diffusion timestep t. Denote $\boldsymbol{v}_p(\boldsymbol{x}_t)$ and $\boldsymbol{v}_\pi(\boldsymbol{x}_t \mid \boldsymbol{y})$ as their corresponding velocity vector fields. We have the following identity:*

$$\frac{\mathrm{d}}{\mathrm{d}t}D_{\mathrm{KL}}(\pi(\boldsymbol{x}_t \mid \boldsymbol{y})\|p(\boldsymbol{x}_t)) = \mathbb{E}_{\boldsymbol{x}_t\sim\pi(\boldsymbol{x}_t|\boldsymbol{y})}\left[(\boldsymbol{v}_\pi(\boldsymbol{x}_t \mid \boldsymbol{y}) - \boldsymbol{v}_p(\boldsymbol{x}_t))^T\nabla_{\boldsymbol{x}_t}\log\pi(\boldsymbol{y} \mid \boldsymbol{x}_t)\right]. \tag{16}$$

*Proof.*  Simplifying the derivative gives:

$$\frac{\mathrm{d}}{\mathrm{d}t}D_{\mathrm{KL}}(\pi(\boldsymbol{x}_t \mid \boldsymbol{y})\|p(\boldsymbol{x}_t)) = \frac{\mathrm{d}}{\mathrm{d}t}\int\pi(\boldsymbol{x}_t \mid \boldsymbol{y})\left[\log\pi(\boldsymbol{x}_t \mid \boldsymbol{y}) - \log p(\boldsymbol{x}_t)\right]d\boldsymbol{x}_t$$

$$= \int\left[\frac{\mathrm{d}}{\mathrm{d}t}\pi(\boldsymbol{x}_t \mid \boldsymbol{y})\right]\left[\log\pi(\boldsymbol{x}_t \mid \boldsymbol{y}) - \log p(\boldsymbol{x}_t)\right]d\boldsymbol{x}_t$$

$$+ \int\pi(\boldsymbol{x}_t \mid \boldsymbol{y})\frac{\mathrm{d}}{\mathrm{d}t}\left[\log\pi(\boldsymbol{x}_t \mid \boldsymbol{y}) - \log p(\boldsymbol{x}_t)\right]d\boldsymbol{x}_t. \quad \text{(Product rule)}$$

Now we expand the first term. Let $\boldsymbol{v}_p(\boldsymbol{x}_t), \boldsymbol{v}_\pi(\boldsymbol{x}_t \mid \boldsymbol{y})$ be the flows at $p(\boldsymbol{x}_t), \pi(\boldsymbol{x}_t \mid \boldsymbol{y})$ respectively:

$$\int\left[\frac{\mathrm{d}}{\mathrm{d}t}\pi(\boldsymbol{x}_t \mid \boldsymbol{y})\right]\left[\log\pi(\boldsymbol{x}_t \mid \boldsymbol{y}) - \log p(\boldsymbol{x}_t)\right]d\boldsymbol{x}_t$$

$$= -\int\nabla\cdot(\pi(\boldsymbol{x}_t \mid \boldsymbol{y})\boldsymbol{v}_\pi(\boldsymbol{x}_t \mid \boldsymbol{y}))\left[\log\pi(\boldsymbol{x}_t \mid \boldsymbol{y}) - \log p(\boldsymbol{x}_t)\right]d\boldsymbol{x}_t \quad \text{(Continuity equation)}$$

$$= \int\pi(\boldsymbol{x}_t \mid \boldsymbol{y})\boldsymbol{v}_\pi(\boldsymbol{x}_t \mid \boldsymbol{y})^T\nabla_{\boldsymbol{x}_t}\left[\log\pi(\boldsymbol{x}_t \mid \boldsymbol{y}) - \log p(\boldsymbol{x}_t)\right]d\boldsymbol{x}_t \quad \text{(Integration by parts)}$$

$$= \int\pi(\boldsymbol{x}_t \mid \boldsymbol{y})\boldsymbol{v}_\pi(\boldsymbol{x}_t \mid \boldsymbol{y})^T\nabla_{\boldsymbol{x}_t}\log\pi(\boldsymbol{y} \mid \boldsymbol{x}_t)d\boldsymbol{x}_t. \quad \text{(Bayes)}$$

For the second term:

$$\int \pi(\boldsymbol{x}_t \mid \boldsymbol{y}) \frac{\mathrm{d}}{\mathrm{d}t} \left[\log \pi(\boldsymbol{x}_t \mid \boldsymbol{y}) - \log p(\boldsymbol{x}_t)\right] d\boldsymbol{x}_t$$

$$= \mathbb{E}_{\boldsymbol{x}_t \sim \pi(\boldsymbol{x}_t|\boldsymbol{y})} \left[\frac{\mathrm{d}}{\mathrm{d}t} \log \pi(\boldsymbol{x}_t \mid \boldsymbol{y})\right] - \int \pi(\boldsymbol{x}_t \mid \boldsymbol{y}) \frac{\mathrm{d}}{\mathrm{d}t} \log p(\boldsymbol{x}_t) d\boldsymbol{x}_t$$

$$= -\int \pi(\boldsymbol{x}_t \mid \boldsymbol{y}) \frac{\mathrm{d}}{\mathrm{d}t} \log p(\boldsymbol{x}_t) d\boldsymbol{x}_t$$

$$= \int \pi(\boldsymbol{x}_t \mid \boldsymbol{y}) \left[\nabla \cdot \boldsymbol{v}_p(\boldsymbol{x}_t) + \boldsymbol{v}_p(\boldsymbol{x}_t)^T \nabla_{\boldsymbol{x}_t} \log p(\boldsymbol{x}_t)\right] d\boldsymbol{x}_t \qquad \text{(Log-continuity equation)}$$

$$= \int \pi(\boldsymbol{x}_t \mid \boldsymbol{y}) \left[\nabla \cdot \boldsymbol{v}_p(\boldsymbol{x}_t) + \boldsymbol{v}_p(\boldsymbol{x}_t)^T \nabla_{\boldsymbol{x}_t} \log \pi(\boldsymbol{x}_t \mid \boldsymbol{y})\right] d\boldsymbol{x}_t$$

$$\quad - \int \pi(\boldsymbol{x}_t \mid \boldsymbol{y}) \boldsymbol{v}_p(\boldsymbol{x}_t)^T \nabla_{\boldsymbol{x}_t} \log \pi(\boldsymbol{y} \mid \boldsymbol{x}_t) d\boldsymbol{x}_t \qquad \text{(Bayes)}$$

$$= -\int \pi(\boldsymbol{x}_t \mid \boldsymbol{y}) \boldsymbol{v}_p(\boldsymbol{x}_t)^T \nabla_{\boldsymbol{x}_t} \log \pi(\boldsymbol{y} \mid \boldsymbol{x}_t) d\boldsymbol{x}_t. \qquad \text{(Stein's identity)}$$

Adding the two terms back together gives us:

$$\frac{\mathrm{d}}{\mathrm{d}t} D_{\mathrm{KL}}(\pi(\boldsymbol{x}_t \mid \boldsymbol{y}) \| p(\boldsymbol{x}_t)) = \int \pi(\boldsymbol{x}_t \mid \boldsymbol{y}) \left(\boldsymbol{v}_\pi(\boldsymbol{x}_t \mid \boldsymbol{y}) - \boldsymbol{v}_p(\boldsymbol{x}_t)\right)^T \nabla_{\boldsymbol{x}_t} \log \pi(\boldsymbol{y} \mid \boldsymbol{x}_t) d\boldsymbol{x}_t$$

$$= \mathbb{E}_{\boldsymbol{x}_t \sim \pi(\boldsymbol{x}_t|\boldsymbol{y})} \left[\left(\boldsymbol{v}_\pi(\boldsymbol{x}_t \mid \boldsymbol{y}) - \boldsymbol{v}_p(\boldsymbol{x}_t)\right)^T \nabla_{\boldsymbol{x}_t} \log \pi(\boldsymbol{y} \mid \boldsymbol{x}_t)\right].$$

$$\square$$

**Proposition 1** (`DiME`: Model evidence estimation along the standard marginals)**.** *Given diffusion process* $\boldsymbol{x}_t = a_t \boldsymbol{x}_0 + \sigma_t \boldsymbol{z}_t$, $\boldsymbol{z}_t \sim \mathcal{N}(0, \boldsymbol{I})$, *posterior marginals* $p(\boldsymbol{x}_t \mid \boldsymbol{y}) \propto p(\boldsymbol{x}_t) p(\boldsymbol{y} \mid \boldsymbol{x}_t)$, *and timesteps* $0 = t_0 < \cdots < t_N = T$, *the model evidence can be estimated via:*

$$\log p(\boldsymbol{y}) = \mathbb{E}_{\boldsymbol{x}_0 \sim p(\boldsymbol{x}_0|\boldsymbol{y})} \left[\log p(\boldsymbol{y} \mid \boldsymbol{x}_0)\right] - D_{\mathrm{KL}}(p(\boldsymbol{x}_0 \mid \boldsymbol{y}) \| p(\boldsymbol{x}_0)). \tag{8}$$

*The log-likelihood term can be estimated with posterior samples* [3]*, and the KL divergence with:*

$$D_{\mathrm{KL}}(p(\boldsymbol{x}_0 \mid \boldsymbol{y}) \| p(\boldsymbol{x}_0)) \approx \sum_{i=1}^{N} c_{t_i} \Delta t_i \mathbb{E}_{\boldsymbol{x}_{t_i} \sim p(\boldsymbol{x}_{t_i}|\boldsymbol{y})} \| \nabla_{\boldsymbol{x}_{t_i}} \log p(\boldsymbol{y} \mid \boldsymbol{x}_{t_i}) \|^2 \tag{9}$$

*where* $c_{t_i} = \sigma'_{t_i} \sigma_{t_i} - \sigma_{t_i}^2 \frac{a'_{t_i}}{a_{t_i}}$ *and* $\Delta t_i = t_i - t_{i-1}$ *are derived from the diffusion schedule.*

*Proof.* Expanding out the KL divergence gives us an expression for the model evidence:

$$D_{\mathrm{KL}}(p(\boldsymbol{x}_0 \mid \boldsymbol{y}) \| p(\boldsymbol{x}_0)) = \mathbb{E}_{\boldsymbol{x}_0 \sim p(\boldsymbol{x}_0|\boldsymbol{y})} \left[\log p(\boldsymbol{x}_0 \mid \boldsymbol{y}) - \log p(\boldsymbol{x}_0)\right]$$

$$= \mathbb{E}_{\boldsymbol{x}_0 \sim p(\boldsymbol{x}_0|\boldsymbol{y})} \left[\log p(\boldsymbol{y} \mid \boldsymbol{x}_0) - \log p(\boldsymbol{y})\right]$$

$$= \mathbb{E}_{\boldsymbol{x}_0 \sim p(\boldsymbol{x}_0|\boldsymbol{y})} \left[\log p(\boldsymbol{y} \mid \boldsymbol{x}_0)\right] - \log p(\boldsymbol{y})$$

$$\implies \log p(\boldsymbol{y}) = \mathbb{E}_{\boldsymbol{x}_0 \sim p(\boldsymbol{x}_0|\boldsymbol{y})} \left[\log p(\boldsymbol{y} \mid \boldsymbol{x}_0)\right] - D_{\mathrm{KL}}(p(\boldsymbol{x}_0 \mid \boldsymbol{y}) \| p(\boldsymbol{x}_0)).$$

---

[3]For likelihoods with Gaussian noise, if the posterior and prior have enough overlap such that posterior samples have a data fit of $\chi^2 \approx 1$, this term essentially becomes constant under the same likelihood function.

At the final diffusion step, the forward process has destroyed all information about the measurements, so we have $p(\boldsymbol{x}_T \mid \boldsymbol{y}) = p(\boldsymbol{x}_T)$, or $D_{\mathrm{KL}}(p(\boldsymbol{x}_T \mid \boldsymbol{y})\|p(\boldsymbol{x}_T)) = 0$. Therefore, we have:

$$
\begin{aligned}
D_{\mathrm{KL}}(p(\boldsymbol{x}_0 \mid \boldsymbol{y})\|p(\boldsymbol{x}_0)) &= -\int_0^T \frac{\mathrm{d}}{\mathrm{d}t} D_{\mathrm{KL}}(p(\boldsymbol{x}_t \mid \boldsymbol{y})\|p(\boldsymbol{x}_t))dt \\
&\overset{(i)}{=} -\int_0^T \mathbb{E}_{\boldsymbol{x}_t \sim p(\boldsymbol{x}_t|\boldsymbol{y})}\left[\left(\boldsymbol{v}_p(\boldsymbol{x}_t \mid \boldsymbol{y}) - \boldsymbol{v}_p(\boldsymbol{x}_t)\right)^\top \nabla_{\boldsymbol{x}_t} \log p(\boldsymbol{y} \mid \boldsymbol{x}_t)\right] dt \\
&\overset{(ii)}{=} \int_0^T \left(\sigma_t'\sigma_t - \sigma_t^2 \frac{a_t'}{a_t}\right) \mathbb{E}_{\boldsymbol{x}_t \sim p(\boldsymbol{x}_t|\boldsymbol{y})}\left\|\nabla_{\boldsymbol{x}_t} \log p(\boldsymbol{y} \mid \boldsymbol{x}_t)\right\|^2 dt. \\
&\approx \sum_{i=1}^N \Delta t_i \left(\sigma_{t_i}'\sigma_{t_i} - \sigma_{t_i}^2 \frac{a_{t_i}'}{a_{t_i}}\right) \mathbb{E}_{\boldsymbol{x}_{t_i} \sim p(\boldsymbol{x}_{t_i}|\boldsymbol{y})}\|\nabla_{\boldsymbol{x}_{t_i}} \log p(\boldsymbol{y} \mid \boldsymbol{x}_{t_i})\|^2.
\end{aligned}
$$

where (i) uses Lemma 3 but with $\pi(\boldsymbol{x}_t \mid \boldsymbol{y}) = p(\boldsymbol{x}_t \mid \boldsymbol{y})$ and (ii) uses the PF-ODE of the conditional flow (Eq. 15). $\square$

A.1 PROOFS AND DERIVATIONS FOR THE DAPS-BASED IMPLEMENTATION

**Lemma 1.** *Suppose we have diffusion process* $\boldsymbol{x}_t = a_t \boldsymbol{x}_0 + \sigma_t \boldsymbol{z}_t, \boldsymbol{z}_t \sim \mathcal{N}(0, \boldsymbol{I})$. *Let* $\boldsymbol{x}_t \sim p(\boldsymbol{x}_t)$ *and suppose* $p(\boldsymbol{x}_0) \approx \mathcal{N}(\boldsymbol{\mu}_0, \boldsymbol{\Sigma}_0)$. *Then we can express the posterior covariance* $\boldsymbol{\Sigma}_{\boldsymbol{x}_0 | \boldsymbol{x}_t}$ *as*

$$\boldsymbol{\Sigma}_{\boldsymbol{x}_0 | \boldsymbol{x}_t} = \left[ \boldsymbol{\Sigma}_0^{-1} + \frac{a_t^2}{\sigma_t^2} \boldsymbol{I} \right]^{-1}. \tag{10}$$

*Proof.* Our diffusion corruption process gives us:

$$\mathrm{Cov}(\boldsymbol{x}_t) = a_t^2 \boldsymbol{\Sigma}_0 + \sigma_t^2 \boldsymbol{I}$$
$$\mathrm{Cov}(\boldsymbol{x}_0, \boldsymbol{x}_t) = \mathrm{Cov}(\boldsymbol{x}_0, a_t \boldsymbol{x}_0 + \sigma_t \boldsymbol{z}_t) = a_t \mathrm{Cov}(\boldsymbol{x}_0) = a_t \boldsymbol{\Sigma}_0.$$

Plugging these into the Gaussian posterior covariance formula gives:

$$\begin{aligned}
\mathrm{Cov}(\boldsymbol{x}_0 \mid \boldsymbol{x}_t) &= \mathrm{Cov}(\boldsymbol{x}_0) - \mathrm{Cov}(\boldsymbol{x}_0, \boldsymbol{x}_t) \mathrm{Cov}(\boldsymbol{x}_t)^{-1} \mathrm{Cov}(\boldsymbol{x}_0, \boldsymbol{x}_t) \\
&= \boldsymbol{\Sigma}_0 - a_t \boldsymbol{\Sigma}_0 \left[ a_t^2 \boldsymbol{\Sigma}_0 + \sigma_t^2 \boldsymbol{I} \right]^{-1} a_t \boldsymbol{\Sigma}_0 \\
&= \boldsymbol{\Sigma}_0 - \boldsymbol{\Sigma}_0 \left[ \boldsymbol{\Sigma}_0 + \frac{\sigma_t^2}{a_t^2} \boldsymbol{I} \right]^{-1} \boldsymbol{\Sigma}_0 \\
&= \left[ \boldsymbol{\Sigma}_0^{-1} + \frac{a_t^2}{\sigma_t^2} \boldsymbol{I} \right]^{-1}. \qquad \text{(Woodbury identity).}
\end{aligned}$$

$\square$

**Corollary 1.** *Suppose the training data has* $\lambda_{max}(\boldsymbol{\Sigma}_0) = 1$. *Then* $\|\boldsymbol{\Sigma}_{\boldsymbol{x}_0 | \boldsymbol{x}_t}\| = \frac{\sigma_t^2}{\sigma_t^2 + a_t^2}$.

*Proof.* Plugging in the covariance from Lemma 1 gives:

$$\|\boldsymbol{\Sigma}_{\boldsymbol{x}_0 | \boldsymbol{x}_t}\| = \lambda_{\max} \left[ \boldsymbol{\Sigma}_0^{-1} + \frac{a_t^2}{\sigma_t^2} \boldsymbol{I} \right]^{-1} = \frac{1}{\lambda_{\min} \left( \boldsymbol{\Sigma}_0^{-1} + \frac{a_t^2}{\sigma_t^2} \boldsymbol{I} \right)} = \frac{1}{1 + \frac{a_t^2}{\sigma_t^2}} = \frac{\sigma_t^2}{\sigma_t^2 + a_t^2}.$$

$\square$

**Lemma 2.** *Suppose we have diffusion process* $\boldsymbol{x}_t = a_t \boldsymbol{x}_0 + \sigma_t \boldsymbol{z}_t, \boldsymbol{z}_t \sim \mathcal{N}(0, \boldsymbol{I})$ *and sample* $\tilde{\boldsymbol{x}}_0 \sim p(\boldsymbol{x}_0 \mid \boldsymbol{x}_t, \boldsymbol{y})$. *Under the assumption that* $p(\boldsymbol{x}_0 \mid \boldsymbol{x}_t) \approx \mathcal{N}(\mathbb{E}[\boldsymbol{x}_0 \mid \boldsymbol{x}_t], \boldsymbol{\Sigma}_{\boldsymbol{x}_0 | \boldsymbol{x}_t})$, *both* $\Theta_{high}(\tilde{\boldsymbol{x}}_0)$ *and* $\Theta_{low}(\tilde{\boldsymbol{x}}_0)$ *are unbiased estimators of* $\nabla_{\boldsymbol{x}_t} \log p(\boldsymbol{y} \mid \boldsymbol{x}_t)$ *with*

$$\Theta_{high}(\tilde{\boldsymbol{x}}_0) = \frac{a_t}{\sigma_t^2} \left( \tilde{\boldsymbol{x}}_0 - \mathbb{E}[\boldsymbol{x}_0 \mid \boldsymbol{x}_t] \right). \tag{11}$$

$$\Theta_{low}(\tilde{\boldsymbol{x}}_0) = \frac{a_t}{\sigma_t^2} \left( \boldsymbol{\Sigma}_{\boldsymbol{x}_0 | \boldsymbol{x}_t} \nabla_{\tilde{\boldsymbol{x}}_0} \log p(\boldsymbol{y} \mid \tilde{\boldsymbol{x}}_0) \right). \tag{12}$$

*Under low diffusion noise* $(a_t \to 1, \sigma_t \to 0)$, $\mathrm{Var}(\Theta_{low}) \to 0$ *but* $\mathrm{Var}(\Theta_{high}) = O(\frac{1}{\sigma_t^2})$.

*Under high noise* $(a_t \to 0, \sigma_t \to 1)$, $\mathrm{Var}(\Theta_{high}) \to 0$ *but* $\mathrm{Var}(\Theta_{low}) = O(\frac{\sigma_t^4}{\sigma_y^4} \mathrm{Var}(\Theta_{high}))$.

*Proof.* Using Tweedie's rule, we can show that $\Theta_{high}$ is unbiased for the likelihood score:

$$\begin{aligned}
\nabla_{\boldsymbol{x}_t} \log p(\boldsymbol{y} \mid \boldsymbol{x}_t) &= \nabla_{\boldsymbol{x}_t} \log p(\boldsymbol{x}_t \mid \boldsymbol{y}) - \nabla_{\boldsymbol{x}_t} \log p(\boldsymbol{x}_t) \\
&= \frac{a_t}{\sigma_t^2} \left( \mathbb{E}[\boldsymbol{x}_0 \mid \boldsymbol{x}_t, \boldsymbol{y}] - \frac{\boldsymbol{x}_t}{a_t} \right) - \frac{a_t}{\sigma_t^2} \left( \mathbb{E}[\boldsymbol{x}_0 \mid \boldsymbol{x}_t] - \frac{\boldsymbol{x}_t}{a_t} \right) \\
&= \frac{a_t}{\sigma_t^2} \left( \mathbb{E}[\boldsymbol{x}_0 \mid \boldsymbol{x}_t, \boldsymbol{y}] - \mathbb{E}[\boldsymbol{x}_0 \mid \boldsymbol{x}_t] \right). \qquad (\star)
\end{aligned}$$

Our Gaussian approximation of $p(\boldsymbol{x}_0 \mid \boldsymbol{x}_t)$ gives us $\nabla_{\boldsymbol{x}_0} \log p(\boldsymbol{x}_0 \mid \boldsymbol{x}_t) = -\boldsymbol{\Sigma}_{\boldsymbol{x}_0|\boldsymbol{x}_t}^{-1}(\boldsymbol{x}_0 - \mathbb{E}[\boldsymbol{x}_0 \mid \boldsymbol{x}_t])$. Using the fact that the expectation of the score is zero, we have:

$$
\begin{aligned}
\mathbb{E}[\boldsymbol{x}_0 \mid \boldsymbol{x}_t, \boldsymbol{y}] &= \mathbb{E}[\boldsymbol{x}_0 + \boldsymbol{\Sigma}_{\boldsymbol{x}_0|\boldsymbol{x}_t} \nabla_{\boldsymbol{x}_0} \log p(\boldsymbol{x}_0 \mid \boldsymbol{x}_t, \boldsymbol{y}) \mid \boldsymbol{x}_t, \boldsymbol{y}] \\
&= \mathbb{E}[\boldsymbol{x}_0 + \boldsymbol{\Sigma}_{\boldsymbol{x}_0|\boldsymbol{x}_t} (\nabla_{\boldsymbol{x}_0} \log p(\boldsymbol{x}_0 \mid \boldsymbol{x}_t) + \nabla_{\boldsymbol{x}_0} \log p(\boldsymbol{y} \mid \boldsymbol{x}_0)) \mid \boldsymbol{x}_t, \boldsymbol{y}] \\
&= \mathbb{E}[\boldsymbol{x}_0 - (\boldsymbol{x}_0 - \mathbb{E}[\boldsymbol{x}_0 \mid \boldsymbol{x}_t]) + \boldsymbol{\Sigma}_{\boldsymbol{x}_0|\boldsymbol{x}_t} \nabla_{\boldsymbol{x}_0} \log p(\boldsymbol{y} \mid \boldsymbol{x}_0) \mid \boldsymbol{x}_t, \boldsymbol{y}] \\
&= \mathbb{E}[\boldsymbol{x}_0 \mid \boldsymbol{x}_t] + \boldsymbol{\Sigma}_{\boldsymbol{x}_0|\boldsymbol{x}_t} \mathbb{E}_{\boldsymbol{x}_0 \sim p(\boldsymbol{x}_0|\boldsymbol{x}_t, \boldsymbol{y})}[\nabla_{\boldsymbol{x}_0} \log p(\boldsymbol{y} \mid \boldsymbol{x}_0)]. \quad\quad (\star\star)
\end{aligned}
$$

Substituting $(\star\star)$ into $(\star)$ shows that $\Theta_{low}$ is unbiased. To compute a variance bound for $\Theta_{high}$,

$$
\begin{aligned}
\mathrm{Var}(\Theta_{high}) &= \frac{a_t^2}{\sigma_t^4} \mathrm{Cov}(\boldsymbol{x}_0 \mid \boldsymbol{x}_t, \boldsymbol{y}) \\
&\leq \frac{a_t^2}{\sigma_t^4} \mathrm{Cov}(\boldsymbol{x}_0 \mid \boldsymbol{x}_t) \quad\quad \text{(Gaussian conditional covariance)} \\
&\leq \frac{a_t^2}{\sigma_t^2(\sigma_t^2 + a_t^2)}. \quad\quad \text{(Corollary 1)}
\end{aligned}
$$

The term $\sigma_t^2 + a_t^2$ is of constant order over the diffusion process, resulting in a variance of $a_t^2/\sigma_t^2$. To compute a bound for $\Theta_{low}$, we linearize the likelihood $f$ at a fixed $\bar{x}$ with $\boldsymbol{A} = \frac{\partial f}{\partial \boldsymbol{x}}\big|_{\bar{\boldsymbol{x}}}$ and get

$$
\begin{aligned}
\mathrm{Var}(\Theta_{low}) &= \frac{a_t^2}{\sigma_t^4} \boldsymbol{\Sigma}_{\boldsymbol{x}_0|\boldsymbol{x}_t} \mathrm{Cov}(\nabla_{\boldsymbol{x}_0} \log p(\boldsymbol{y} \mid \boldsymbol{x}_0) \mid \boldsymbol{x}_t, \boldsymbol{y}) \boldsymbol{\Sigma}_{\boldsymbol{x}_0|\boldsymbol{x}_t}^T \\
&= \frac{a_t^2}{\sigma_t^4} \boldsymbol{\Sigma}_{\boldsymbol{x}_0|\boldsymbol{x}_t} \mathrm{Cov}(\frac{1}{\sigma_{\boldsymbol{y}}^2} \boldsymbol{A}^T(\boldsymbol{y} - \boldsymbol{A}\boldsymbol{x}_0) \mid \boldsymbol{x}_t, \boldsymbol{y}) \boldsymbol{\Sigma}_{\boldsymbol{x}_0|\boldsymbol{x}_t}^T \\
&= \frac{1}{\sigma_{\boldsymbol{y}}^4} \boldsymbol{\Sigma}_{\boldsymbol{x}_0|\boldsymbol{x}_t} \boldsymbol{A}^T \boldsymbol{A} \left( \frac{a_t^2}{\sigma_t^4} \mathrm{Cov}(\boldsymbol{x}_0 \mid \boldsymbol{x}_t, \boldsymbol{y}) \right) \boldsymbol{A}^T \boldsymbol{A} \boldsymbol{\Sigma}_{\boldsymbol{x}_0|\boldsymbol{x}_t}^T \\
&\leq \frac{\|\boldsymbol{A}\|^4}{\sigma_{\boldsymbol{y}}^4} \frac{\sigma_t^4}{(\sigma_t^2 + a_t^2)^2} \mathrm{Var}(\Theta_{high}). \quad\quad \text{(Corollary 1)} \\
&\leq \frac{\|\boldsymbol{A}\|^4}{\sigma_{\boldsymbol{y}}^4} \frac{\sigma_t^2 a_t^2}{(\sigma_t^2 + a_t^2)^3}.
\end{aligned}
$$

Unlike $\Theta_{high}$, the variance of this estimator goes to $0$ as $\sigma_t \to 0$, but for the rest of diffusion process the constant factor makes it a higher variance estimator than $\Theta_{high}$ in practice.

$\square$

## B  GENERALIZING DiME TO ARBITRARY POSTERIOR MARGINALS

While we only present DiME along the standard marginals in the main text, DiME can be extended to any arbitrary diffusion posterior marginals that anneal to the true posterior as follows:

**Corollary 2** (Generalized DiME). *Let $p(\boldsymbol{x}_t)$ be the unconditional diffusion marginal and $\pi(\boldsymbol{x}_t \mid \boldsymbol{y}) \propto p(\boldsymbol{x}_t)\pi(\boldsymbol{y} \mid \boldsymbol{x}_t)$ be a path of marginals that anneals to the true posterior, or $\pi(\boldsymbol{x}_0 \mid \boldsymbol{y}) = p(\boldsymbol{x}_0 \mid \boldsymbol{y})$. Denote $\boldsymbol{v}_p(\boldsymbol{x}_t)$ and $\boldsymbol{v}_\pi(\boldsymbol{x}_t \mid \boldsymbol{y})$ as their corresponding velocity fields. Given reverse diffusion timesteps $0 = t_0 < \cdots < t_N = T$, we have:*

$$\log p(\boldsymbol{y}) = \mathbb{E}_{\boldsymbol{x}_0 \sim p(\boldsymbol{x}_0|\boldsymbol{y})}\big[ \log p(\boldsymbol{y} \mid \boldsymbol{x}_0) \big] - D_{\mathrm{KL}}(p(\boldsymbol{x}_0 \mid \boldsymbol{y})\|p(\boldsymbol{x}_0))$$

*where*

$$D_{\mathrm{KL}}(p(\boldsymbol{x}_0 \mid \boldsymbol{y})\|p(\boldsymbol{x}_0)) \approx -\sum_{i=1}^{N} \Delta t_i \mathbb{E}_{\boldsymbol{x}_t \sim \pi(\boldsymbol{x}_t|\boldsymbol{y})}\Big[ \big(\boldsymbol{v}_\pi(\boldsymbol{x}_t \mid \boldsymbol{y}) - \boldsymbol{v}_p(\boldsymbol{x}_t)\big)^\top \nabla_{\boldsymbol{x}_t} \log \pi(\boldsymbol{y} \mid \boldsymbol{x}_t) \Big]$$
$$+ D_{\mathrm{KL}}(\pi(\boldsymbol{x}_T \mid \boldsymbol{y})\|p(\boldsymbol{x}_T)).$$

*Proof.* Follows directly from the proof of Prop. 1.  □

While $\boldsymbol{v}_p$ can be directly computed using the PF-ODE (Eq. 13), computing $\boldsymbol{v}_\pi$ can be difficult for arbitrary time-marginals $\pi$. By crafting a sequence of diffusion marginals with known velocity, we can avoid any approximation like the one used for $p(\boldsymbol{x}_0 \mid \boldsymbol{x}_t)$ in the DAPS implementation. The marginals $\pi(\boldsymbol{x}_t \mid \boldsymbol{y})$ should also ideally have high overlap with the unconditional marginal $p(\boldsymbol{x}_t)$ to ensure accurate computation of $\boldsymbol{v}_p(\boldsymbol{x}_t)$ when the score is learned. If $\pi(\boldsymbol{x}_T \mid \boldsymbol{y}) \neq p(\boldsymbol{x}_T)$, then we must use other estimation methods to compute $D_{\mathrm{KL}}(\pi(\boldsymbol{x}_T \mid \boldsymbol{y})\|p(\boldsymbol{x}_T))$ or settle for an unnormalized model evidence estimate. The following section discusses two estimators that does not require any approximation, but are not normalized.

### B.1  ESTIMATING THE MODEL EVIDENCE FROM ALTERNATIVE SAMPLING METHODS

---

**Algorithm 2** One posterior sample-path of DiME-PnPDM

---

**Require:** Score-based model $s_\theta$, measurement $\boldsymbol{y}$, noise schedule $\sigma_t$, $(t_i)_{i \in \{1,\dots,N\}}$,
 1: Sample $\hat{\boldsymbol{x}}_0 \sim p(\boldsymbol{x}_0)$.
 2: Initialize the integral sum, $f = 0$.
 3: **for** $i = N, N-1 \dots, 0$ **do**
 4:   Sample $\boldsymbol{x}_{t_i} \sim q(\boldsymbol{x}_{t_i} \mid \boldsymbol{y}, \hat{\boldsymbol{x}}_0) \propto q(\boldsymbol{y} \mid \boldsymbol{x}_{t_i})p(\boldsymbol{x}_{t_i} \mid \hat{\boldsymbol{x}}_0)$.
 5:   Sample $\hat{\boldsymbol{x}}_0 \sim p(\boldsymbol{x}_0 \mid \boldsymbol{x}_{t_i})$ using $s_\theta$.
 6:   Update the integral sum $f \leftarrow f - (t_i - t_{i-1})\boldsymbol{v}_p(\boldsymbol{x}_{t_i})^T \nabla_{\boldsymbol{x}_t} \log q(\boldsymbol{y} \mid \boldsymbol{x}_{t_i})$
 7: **end for**
 8: **return** $x^{(0)}$, $f$

---

The PnP-DM marginals follow $q(\boldsymbol{x}_t \mid \boldsymbol{y}) \propto q(\boldsymbol{y} \mid \boldsymbol{x}_t)p(\boldsymbol{x}_t)$ where $q(\boldsymbol{y} \mid \boldsymbol{x}_t)$ is a time-independent likelihood term. As a result, the initial distribution $q(\boldsymbol{x}_T \mid \boldsymbol{y}) \neq p(\boldsymbol{x}_T)$ and the KL divergence $D_{\mathrm{KL}}(q(\boldsymbol{x}_T \mid \boldsymbol{y})\|p(\boldsymbol{x}_T))$ is non-zero, so we can only compute the unnormalized model evidence. However, the unknown constant is dependent on $\boldsymbol{y}$ and the forward model, so this unnormalized quantity can still be used for model selection. We derive the estimator in Proposition 2 and implement it with PnP-DM in Algorithm 2.

**Proposition 2** (DiME-PnPDM: model evidence estimation along the PnP-DM marginals). *Given diffusion process $\boldsymbol{x}_t = a_t\boldsymbol{x}_0 + \sigma_t\boldsymbol{z}_t, \boldsymbol{z}_t \sim \mathcal{N}(0, \boldsymbol{I})$, PnP-DM marginals $q(\boldsymbol{x}_t \mid \boldsymbol{y}) \propto p(\boldsymbol{x}_t)q(\boldsymbol{y} \mid \boldsymbol{x}_t)$, and timesteps $0 = t_0 < \cdots < t_N = T$, the model evidence can be estimated via:*

$$\log p(\boldsymbol{y}) \approx C(q, \boldsymbol{y}) - \sum_{i=1}^{N} \Delta t_i \mathbb{E}_{\boldsymbol{x} \sim q(\boldsymbol{x}_{t_i}|\boldsymbol{y})} \big[ \boldsymbol{v}_p(\boldsymbol{x}_{t_i})^T \nabla_{\boldsymbol{x}_{t_i}} \log q(\boldsymbol{y} \mid \boldsymbol{x}_{t_i}) \big] \qquad (17)$$

*where $C(q, \boldsymbol{y}) = \log \mathbb{E}_{\boldsymbol{x} \sim \mathcal{N}(0, \boldsymbol{I})}[q(\boldsymbol{y} \mid \boldsymbol{x})]$ is constant for fixed measurement $\boldsymbol{y}$ and forward model, and $\boldsymbol{v}_p(\boldsymbol{x}_t) = \frac{a'_t}{a_t}\boldsymbol{x}_t - (\sigma'_t\sigma_t - \sigma_t^2\frac{a'_t}{a_t})\nabla_{\boldsymbol{x}} \log p(\boldsymbol{x}_t)$ is the PF-ODE velocity at $\boldsymbol{x}_t$ (Eq. 13).*

*Proof.* While we can apply Corollary 2, we can take a shortcut to cancel some extra terms. Under the PnP-DM marginals, $\log q(\boldsymbol{y} \mid t)$ is time-dependent while the likelihood $q(\boldsymbol{y} \mid \boldsymbol{x}_t)$ is time-independent, so we can directly compute the derivative of the evidence:

$$
\begin{aligned}
\frac{d}{dt} \log q(\boldsymbol{y} \mid t) &= \mathbb{E}_{\boldsymbol{x} \sim q(\boldsymbol{x}_t \mid \boldsymbol{y})} \left[ \frac{d}{dt} \log p(\boldsymbol{x}_t) \right] && \text{(Fisher's identity)} \\
&= -\mathbb{E}_{\boldsymbol{x} \sim q(\boldsymbol{x}_t \mid \boldsymbol{y})} \left[ \nabla \cdot \boldsymbol{v}_p(\boldsymbol{x}_t) + \boldsymbol{v}_p(\boldsymbol{x}_t)^T \nabla_{\boldsymbol{x}_t} \log p(\boldsymbol{x}_t) \right] && \text{(Log-cont. equation)} \\
&= -\mathbb{E}_{\boldsymbol{x} \sim q(\boldsymbol{x}_t \mid \boldsymbol{y})} \left[ \nabla \cdot \boldsymbol{v}_p(\boldsymbol{x}_t) + \boldsymbol{v}_p(\boldsymbol{x}_t)^T \nabla_{\boldsymbol{x}_t} \log q(\boldsymbol{x}_t \mid \boldsymbol{y}) \right] \\
&\quad + \mathbb{E}_{\boldsymbol{x} \sim q(\boldsymbol{x}_t \mid \boldsymbol{y})} \left[ \boldsymbol{v}_p(\boldsymbol{x}_t)^T \nabla_{\boldsymbol{x}_t} \log q(\boldsymbol{y} \mid \boldsymbol{x}_t) \right] && \text{(Bayes)} \\
&= \mathbb{E}_{\boldsymbol{x} \sim q(\boldsymbol{x}_t \mid \boldsymbol{y})} \left[ \boldsymbol{v}_p(\boldsymbol{x}_t)^T \nabla_{\boldsymbol{x}_t} \log q(\boldsymbol{y} \mid \boldsymbol{x}_t) \right] . && \text{(Stein's identity)}
\end{aligned}
$$

Integrating over all $t$ gives us the estimator:

$$
\begin{aligned}
\log p(\boldsymbol{y}) &= \log q(\boldsymbol{y} \mid t = 0) \\
&= \log q(\boldsymbol{y} \mid t = T) - \int_0^T \frac{d}{dt} \log q(\boldsymbol{y} \mid t) dt \\
&= \log q(\boldsymbol{y} \mid t = T) - \sum_{i=1}^N \Delta t_i \mathbb{E}_{\boldsymbol{x} \sim q(\boldsymbol{x}_{t_i} \mid \boldsymbol{y})} \left[ \boldsymbol{v}_p(\boldsymbol{x}_{t_i})^T \nabla_{\boldsymbol{x}_{t_i}} \log q(\boldsymbol{y} \mid \boldsymbol{x}_{t_i}) \right]
\end{aligned}
$$

where $\log q(\boldsymbol{y} \mid t = T) = \log \mathbb{E}_{\boldsymbol{x} \sim \mathcal{N}(0,\sigma_T^2)} [q(\boldsymbol{y} \mid \boldsymbol{x})]$ is the model evidence under a Gaussian prior, and $\boldsymbol{v}_p(\boldsymbol{x}_t) = \frac{a_t'}{a_t} \boldsymbol{x}_t - (\sigma_t' \sigma_t - \sigma_t^2 \frac{a_t'}{a_t}) \nabla_{\boldsymbol{x}} \log p(\boldsymbol{x}_t)$ is the PF-ODE from Equation 13.

$\square$

We can also do a similar derivation for the marginals used for the Twisted Diffusion Sampler (Wu et al., 2023), which follow $\pi(\boldsymbol{x}_t \mid \boldsymbol{y}) \propto p(\boldsymbol{x}_t) p(\boldsymbol{y} \mid \mathbb{E}[\boldsymbol{x}_0 \mid \boldsymbol{x}_t])$; note that this evidence gradient can be computed before resampling using SMC weights or after resampling without weights.

**Proposition 3** (`DiME-TDS`: model evidence estimation along the TDS marginals). *Given diffusion process $\boldsymbol{x}_t = a_t \boldsymbol{x}_0 + \sigma_t \boldsymbol{z}_t, \boldsymbol{z}_t \sim \mathcal{N}(0, \boldsymbol{I})$, TDS marginals $\pi(\boldsymbol{x}_t \mid \boldsymbol{y}) \propto p(\boldsymbol{x}_t) p(\boldsymbol{y} \mid \boldsymbol{\mu}(\boldsymbol{x}_t))$, and timesteps $0 = t_0 < \cdots < t_N = T$, the model evidence can be estimated via:*

$$
\log p(\boldsymbol{y}) \approx C(q, \boldsymbol{y}) - \sum_{i=1}^N \Delta t_i \mathbb{E}_{\boldsymbol{x}_t \sim \pi(\boldsymbol{x}_{t_i} \mid \boldsymbol{y})} \left[ \boldsymbol{v}_p(\boldsymbol{x}_{t_i})^T \nabla_{\boldsymbol{x}_{t_i}} \log p(\boldsymbol{y} \mid \boldsymbol{\mu}(\boldsymbol{x}_{t_i})) + \frac{d}{dt} \log p(\boldsymbol{y} \mid \boldsymbol{\mu}(\boldsymbol{x}_{t_i})) \right]
$$

(18)

*where $C(q, \boldsymbol{y}) = \log \mathbb{E}_{\boldsymbol{x} \sim \mathcal{N}(0,\sigma_T^2)} [p(\boldsymbol{y} \mid \boldsymbol{\mu}(\boldsymbol{x}_T))]$ is constant for fixed measurement $\boldsymbol{y}$ and forward model, and $\boldsymbol{v}_p(\boldsymbol{x}_t) = \frac{a_t'}{a_t} \boldsymbol{x}_t - (\sigma_t' \sigma_t - \sigma_t^2 \frac{a_t'}{a_t}) \nabla_{\boldsymbol{x}} \log p(\boldsymbol{x}_t)$ is the PF-ODE velocity at $\boldsymbol{x}_t$ (Eq. 13).*

*Proof.*

$$
\begin{aligned}
\frac{d}{dt} \log \pi(\boldsymbol{y} \mid t) &= \frac{1}{\pi(\boldsymbol{y} \mid t)} \frac{d}{dt} \int p(\boldsymbol{x}_t) p(\boldsymbol{y} \mid \boldsymbol{\mu}(\boldsymbol{x}_t)) d\boldsymbol{x}_t \\
&= \frac{1}{\pi(\boldsymbol{y} \mid t)} \int \left[ \frac{d}{dt} p(\boldsymbol{x}_t) \right] p(\boldsymbol{y} \mid \boldsymbol{\mu}(\boldsymbol{x}_t)) d\boldsymbol{x}_t + \frac{1}{\pi(\boldsymbol{y} \mid t)} \int p(\boldsymbol{x}_t) \left[ \frac{d}{dt} p(\boldsymbol{y} \mid \boldsymbol{\mu}(\boldsymbol{x}_t)) \right] d\boldsymbol{x}_t \\
&= \mathbb{E}_{\boldsymbol{x}_t \sim \pi(\boldsymbol{x}_t \mid \boldsymbol{y})} \left[ \frac{d}{dt} \log p(\boldsymbol{x}_t) + \frac{d}{dt} \log p(\boldsymbol{y} \mid \boldsymbol{\mu}(\boldsymbol{x}_t)) \right] \\
&= \mathbb{E}_{\boldsymbol{x} \sim \pi(\boldsymbol{x}_t \mid \boldsymbol{y})} \left[ \boldsymbol{v}_p(\boldsymbol{x}_t)^T \nabla_{\boldsymbol{x}_t} \log p(\boldsymbol{y} \mid \boldsymbol{\mu}(\boldsymbol{x}_t)) + \frac{d}{dt} \log p(\boldsymbol{y} \mid \boldsymbol{\mu}(\boldsymbol{x}_t)) \right]
\end{aligned}
$$

where the last line is derived in a similarly to Proposition 2. Integrating over all $t$ gives the estimator:

$$
\begin{aligned}
\log p(\boldsymbol{y}) &= \log \pi(\boldsymbol{y} \mid t = T) - \int_0^T \frac{d}{dt} \log \pi(\boldsymbol{y} \mid t) dt \\
&\approx \log \pi(\boldsymbol{y} \mid t = T) - \sum_{i=1}^N \Delta t_i \mathbb{E}_{\boldsymbol{x}_t \sim \pi(\boldsymbol{x}_{t_i} \mid \boldsymbol{y})} \left[ \boldsymbol{v}_p(\boldsymbol{x}_{t_i})^T \nabla_{\boldsymbol{x}_{t_i}} \log p(\boldsymbol{y} \mid \boldsymbol{\mu}(\boldsymbol{x}_{t_i})) + \frac{d}{dt} \log p(\boldsymbol{y} \mid \boldsymbol{\mu}(\boldsymbol{x}_{t_i})) \right]
\end{aligned}
$$

where $\log \pi(\boldsymbol{y} \mid T) \approx \log \mathbb{E}_{\boldsymbol{x} \sim \mathcal{N}(0, \sigma_T^2)} \left[ p(\boldsymbol{y} \mid \mu(\boldsymbol{x}_T)) \right].$ $\qquad \square$

## C  BASELINE METHODS

In Section 4.1, we compared to five baseline methods, described here. Define $p_\beta$ as an intermediate distribution of the *power-posterior path*, used in TI, AIS, and SMC: $p_\beta(\boldsymbol{x}) \propto p(\boldsymbol{x})p(\boldsymbol{y} \mid \boldsymbol{x})^\beta, \beta \in [0, 1]$. Note that this path is equivalent to the *geometric path* under a different parametrization.

**Naive Monte Carlo (Naive MC)**: We generate many unconditional prior samples and compute the expectation of the likelihood as follows: $\log p(\boldsymbol{y}) = \log \mathbb{E}_{\boldsymbol{x} \sim p(\boldsymbol{x}_0)}[p(\boldsymbol{y} \mid \boldsymbol{x})]$.

**Original DAPS covariance heuristic (Original heuristic)**: We perform the same steps as described in Algorithm 1 but using the originally proposed covariance heuristic $\boldsymbol{\Sigma}_{\boldsymbol{x}_0 \mid \boldsymbol{x}_t} = \sigma_t^2$.

**Thermodynamic integration (TI)** (Lartillot & Philippe, 2006): Originally from statistical mechanics, TI can also compute expectations by integrating the evidence over the power-posterior path:

$$\frac{d}{d\beta} \log p_\beta(\boldsymbol{y}) = \frac{1}{p_\beta(\boldsymbol{y})} \frac{d}{d\beta} p_\beta(\boldsymbol{y}) = \frac{1}{p_\beta(\boldsymbol{y})} \int p(\boldsymbol{x}) \frac{d}{d\beta} p(\boldsymbol{y} \mid \boldsymbol{x})^\beta d\boldsymbol{x}$$

$$= \frac{1}{p_\beta(\boldsymbol{y})} \int p(\boldsymbol{x})p(\boldsymbol{y} \mid \boldsymbol{x})^\beta \log p(\boldsymbol{y} \mid \boldsymbol{x}) d\boldsymbol{x} = \mathbb{E}_{\boldsymbol{x} \sim p_\beta}[\log p(\boldsymbol{y} \mid \boldsymbol{x})].$$

Using the fact that $\log p_{\beta=0}(\boldsymbol{y}) = \log 1 = 0$, using $K$ integration steps we have the estimator:

$$\log p_{\beta=1}(\boldsymbol{y}) \approx \sum_{k=1}^{K} \Delta\beta_k \, \mathbb{E}_{\boldsymbol{x} \sim p_{\beta_{k-1}}}[\log p(\boldsymbol{y} \mid \boldsymbol{x})].$$

Samples from each $p_\beta$ were generated using Sequential Monte Carlo (details below).

**Annealed importance sampling (AIS)** (Neal, 2001): We can compute the ratio of evidence:

$$\frac{p_{\beta_k}(\boldsymbol{y})}{p_{\beta_{k-1}}(\boldsymbol{y})} = \frac{\int p(\boldsymbol{x})p(\boldsymbol{y} \mid \boldsymbol{x})^{\beta_{k-1}} p(\boldsymbol{y} \mid \boldsymbol{x})^{\Delta\beta_k} d\boldsymbol{x}}{p_{\beta_{k-1}}(\boldsymbol{y})} = \mathbb{E}_{\boldsymbol{x} \sim p_{\beta_{k-1}}}[p(\boldsymbol{y} \mid \boldsymbol{x})^{\Delta\beta_k}]$$

$$= \mathbb{E}_{p_{\boldsymbol{x} \sim \beta_{k-1}}}[\exp(\Delta\beta_k \log p(\boldsymbol{y} \mid \boldsymbol{x}))].$$

Using the fact that $\log p_{\beta=0}(\boldsymbol{y}) = \log 1 = 0$, telescoping over all $\beta_k$ gives:

$$p_{\beta=1}(\boldsymbol{y}) = \prod_{k=1}^{K} \mathbb{E}_{p_{\beta_{k-1}}}[\exp(\Delta\beta_k \log p(\boldsymbol{y} \mid \boldsymbol{x}))]$$

$$= \mathbb{E}_{\boldsymbol{x}_{0:k-1} \sim p_{\beta_{0:k-1}}} \exp\left( \sum_{k=1}^{K} \Delta\beta_k \log p(\boldsymbol{y} \mid \boldsymbol{x}_{k-1}) \right).$$

Note that $\boldsymbol{x}_{0:k-1} \sim p_{\beta_{0:k-1}}$ represents the sample-path of one particle, where each transition kernel from $k-1$ to $k$ was performed using Langevin dynamics.

**Sequential Monte Carlo (SMC)** (Del Moral et al., 2006): An extension of AIS that uses the resampling step from Sequential Monte Carlo methods to replace particles with degenerate weights (i.e. particles stuck in the wrong mode) while sampling the next intermediate distribution, potentially reducing both bias and variance. These resampling steps are performed either using a fixed schedule or adaptively when the effective sample size falls below a threshold.

# D  IMPLEMENTATION DETAILS

## D.1  DiME

As `DiME` is implemented alongside DAPS, we have a similar set of hyperparameters.

**Number of diffusion steps for sampling $p(\boldsymbol{x}_0 \mid \boldsymbol{x}_t)$:** While the original DAPS method uses many diffusion steps to estimate the denoised image at each annealing iteration, we use 1 diffusion step to ensure an exact estimate of $\mathbb{E}[\boldsymbol{x}_0 \mid \boldsymbol{x}_t]$ by Tweedie's formula.

**Langevin dynamics:** The learning rate of Langevin dynamics was tuned so that posterior samples $\tilde{\boldsymbol{x}}_0 \sim p(\boldsymbol{x}_0 \mid \boldsymbol{x}_t, \boldsymbol{y})$ have a mean reduced $\chi^2$ data likelihood fit that converges as low as possible (ideally around 1, which is indicative of a good posterior sample for likelihoods with Gaussian noise). We also use a linear learning rate decay as is done in Zhang et al. (2025). The table below lists the learning rates and maximum step counts used for each experiment. In practice, sample convergence usually happens much sooner than the maximum step count.

| Task | lr | Max steps |
|---|---|---|
| Mixture of Gaussians (in-distribution) | $5 \times 10^{-4}$ | 2,000 |
| Mixture of Gaussians (out-of-distribution) | $2 \times 10^{-5}$ | 10,000 |
| Mixture of Gaussians (saddle point) | $2 \times 10^{-5}$ | 10,000 |
| Fourier phase retrieval | $1 \times 10^{-3}$ | 1,000 |
| Gaussian phase retrieval | $5 \times 10^{-4}$ | 1,000 |
| M87* black hole imaging | $1 \times 10^{-5}$ | 1,000 |

**Selection of $\sigma_{min}$:** This parameter can be viewed as the minimum noise level that the learned score can still be trusted. For the analytic Mixture of Gaussians experiment, we use a minimum annealing noise level $\sigma_{min} = 0.05$, and for the diffusion model experiments, we use $\sigma_{min} = 0.1$ as is done in Zhang et al. (2025). The remaining integral from $t = 0$ to $t = t_{min}$ in Eq. 9 is approximated by a trapezoid, using the fact that the integrand equals 0 at $t = 0$.

**Computing the precision matrix $\boldsymbol{\Sigma}_0^{-1}$:** for some datasets, the covariance of specific pixel locations is nearly 0 (such as the corner regions of MNIST), so we added a jitter of 1e-2 for stability.

## D.2  BASELINES

We use TI, AIS, and SMC as baseline methods for the mixture of Gaussians experiment (Section 4.1) and SMC for the MNIST experiment (Section 4.2; Appendix E). We used Langevin dynamics with 2000 steps and learning rate 0.0002 for our transitions between distributions. For SMC, we resample when the effective sample size (ESS) is less than 0.6.

We test with fixed linear, exponential, sigmoidal $\beta$ as well as adaptive $\beta$ schedules that decrease the Conditional ESS at a constant rate of [0.5, 0.6, 0.7]. For the mixture of Gaussians study, an exponential schedule with $\lambda = 6$ was found to be optimal. For the MNIST experiment, the fixed schedules use 50 annealing steps, while the adaptive schedule had a minimum $\beta$ step size of 0.002 which in practice results in 150 steps at most.

# E BASELINE RESULTS ON MNIST MODEL SELECTION

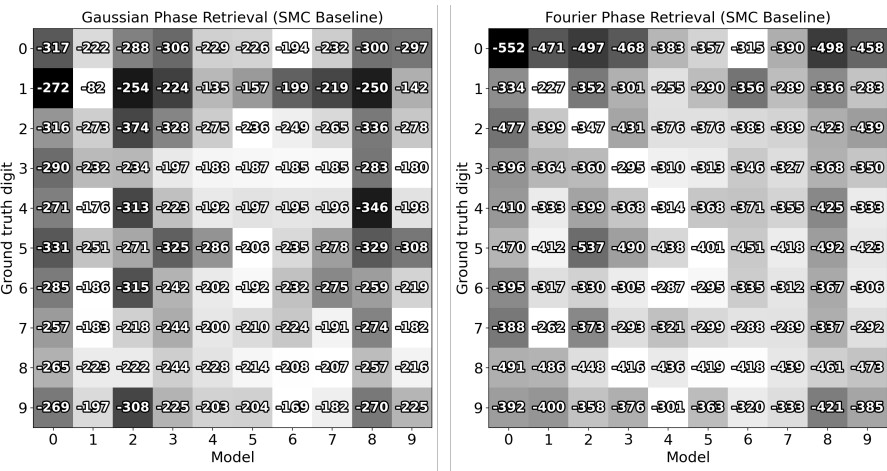

Figure 6: Model evidence confusion matrix using sequential Monte Carlo (SMC) for Gaussian phase retrieval (**left**) and Fourier phase retrieval (**right**) for each (ground truth measurement, model) pair of MNIST digits. When SMC uses the learned diffusion score $\nabla_{\boldsymbol{x}} \log p(\boldsymbol{x}_t)$ at $t = 0.1$ as the prior score, it often fails to select the correct model.

Results of the MNIST experiment (Section 4.2) using SMC and the learned $\nabla_{\boldsymbol{x}} \log p(\boldsymbol{x}_t)$ with $\sigma = 0.1$ as a prior score are shown in Figure 6. SMC is often unable to determine the correct MNIST model. Each SMC annealing step also takes about $10\times$ longer than an equivalent annealing step for `DiME`, due to having to query the diffusion model many times for the prior score. Interestingly, narrower priors (i.e. MNIST digit 1) leads to *higher* model evidence estimates here; since the training data is more clumped together, the learned score becomes more accurate at low noise levels.

For completeness, the model evidence estimates of a few runs over all temperature annealing schedules as well as for a second noise level, $\sigma = 0.05$, are reported in Tables 2 and 3. Mean and standard deviation over 5 runs is shown here. As can be seen, lowering the noise level only further increases overfitting to training data, causing the estimated evidence to drop.

| SMC schedule | $\sigma_{\min} = 0.1$ | | | $\sigma_{\min} = 0.05$ | | |
| --- | --- | --- | --- | --- | --- | --- |
| | GT 8 Model 6 | GT 8 Model 7 | GT 8 Model 8 | GT 8 Model 6 | GT 8 Model 7 | GT 8 Model 8 |
| Linear | $-182 \pm 9$ | $-212 \pm 9$ | $-210 \pm 11$ | $-279 \pm 26$ | $-352 \pm 3$ | $-426 \pm 40$ |
| Exponential | $-183 \pm 7$ | $-224 \pm 5$ | $-239 \pm 6$ | $-292 \pm 20$ | $-383 \pm 9$ | $-468 \pm 12$ |
| Sigmoidal | $-175 \pm 5$ | $-219 \pm 13$ | $-191 \pm 2$ | $-273 \pm 15$ | $-374 \pm 9$ | $-480 \pm 17$ |
| Adaptive 0.5 | $-195 \pm 31$ | $-202 \pm 23$ | $-261 \pm 1$ | $-296 \pm 52$ | $-346 \pm 16$ | $-489 \pm 33$ |
| Adaptive 0.6 | $-184 \pm 6$ | $-189 \pm 7$ | $-259 \pm 4$ | $-279 \pm 45$ | $-373 \pm 17$ | $-485 \pm 26$ |
| Adaptive 0.7 | $-192 \pm 14$ | $-192 \pm 2$ | $-255 \pm 11$ | $-296 \pm 26$ | $-384 \pm 18$ | $-522 \pm 30$ |

Table 2: SMC model evidence estimates on Gaussian phase retrieval.

| SMC schedule | $\sigma_{\min} = 0.1$ | | | $\sigma_{\min} = 0.05$ | | |
| --- | --- | --- | --- | --- | --- | --- |
| | GT 8 Model 6 | GT 8 Model 7 | GT 8 Model 8 | GT 8 Model 6 | GT 8 Model 7 | GT 8 Model 8 |
| Linear | $-416 \pm 11$ | $-431 \pm 4$ | $-409 \pm 1$ | $-485 \pm 25$ | $-577 \pm 2$ | $-563 \pm 2$ |
| Exponential | $-422 \pm 20$ | $-427 \pm 2$ | $-467 \pm 11$ | $-527 \pm 10$ | $-582 \pm 12$ | $-585 \pm 9$ |
| Sigmoidal | $-415 \pm 9$ | $-451 \pm 3$ | $-411 \pm 7$ | $-520 \pm 13$ | $-580 \pm 7$ | $-575 \pm 5$ |
| Adaptive 0.5 | $-409 \pm 4$ | $-409 \pm 6$ | $-460 \pm 8$ | $-524 \pm 33$ | $-563 \pm 18$ | $-567 \pm 10$ |
| Adaptive 0.6 | $-431 \pm 20$ | $-437 \pm 11$ | $-458 \pm 8$ | $-510 \pm 9$ | $-566 \pm 14$ | $-562 \pm 13$ |
| Adaptive 0.7 | $-442 \pm 9$ | $-437 \pm 8$ | $-453 \pm 1$ | $-516 \pm 6$ | $-584 \pm 3$ | $-570 \pm 6$ |

Table 3: SMC model evidence estimates on Fourier phase retrieval.

# F SYNTHETIC MODEL SELECTION

In this section, we perform model selection on the same set of five priors as in Section 4.3.1, but on simulated closure quantities measurements of test images from each prior as verification. The correct model is chosen for all measurements. The posterior samples with the lowest, mean, and highest path evidences are also shown in each case.

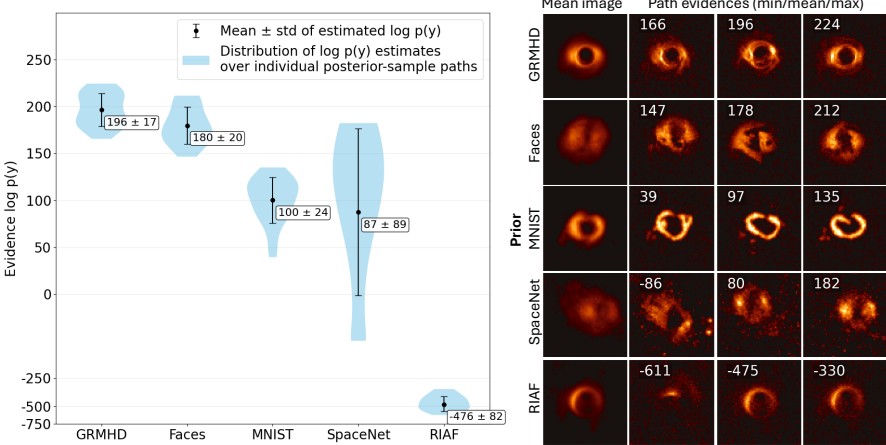

Figure 7: Ground truth: GRMHD.

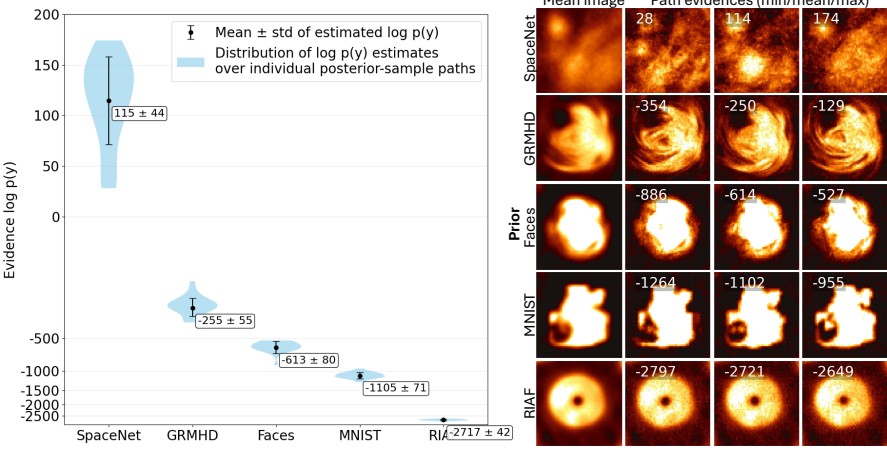

Figure 8: Ground truth: SpaceNet.

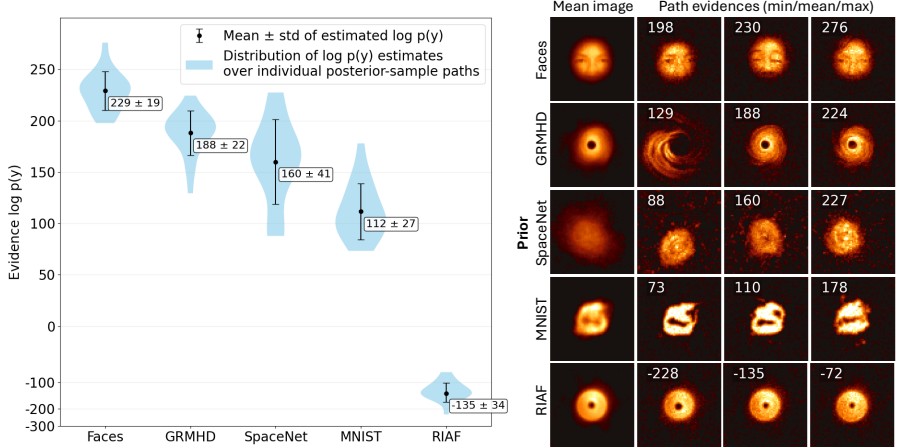

Figure 9: Ground truth: CelebA.

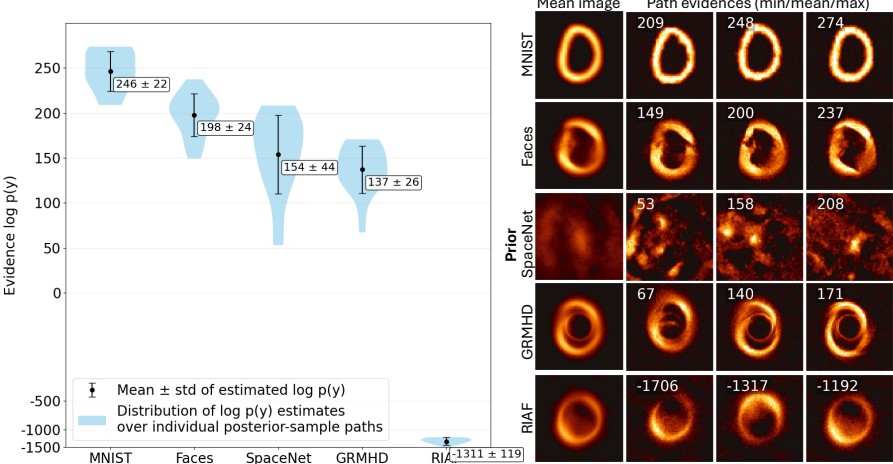

Figure 10: Ground truth: MNIST.

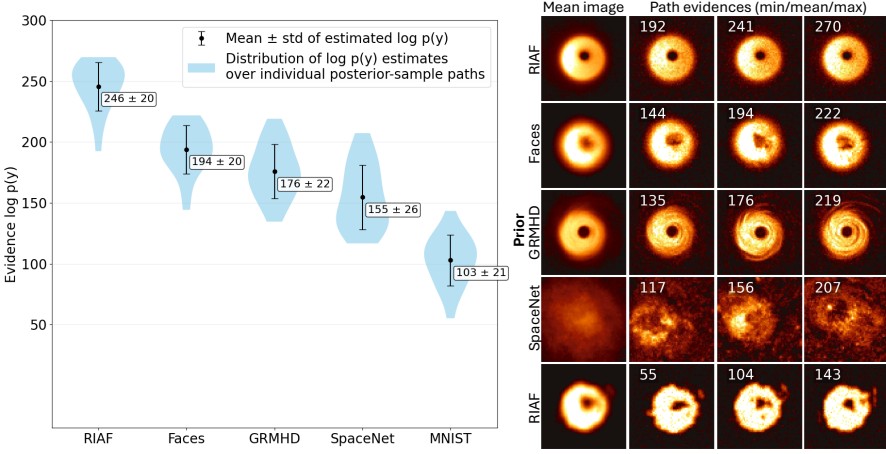

Figure 11: Ground truth: RIAF.

## G  LARGE LANGUAGE MODEL USAGE

We used ChatGPT for coding assistance, polishing text and LaTeX, and checking mathematical derivations.

