# OpenReview forum: "Sample-efficient evidence estimation of score based priors for model selection"
_ICLR.cc/2026/Conference — ICLR 2026 Poster_

### Official Review · Reviewer_56j4 · 2025-10-24

**Soundness:** 3
**Presentation:** 3
**Contribution:** 2
**Rating:** 4
**Confidence:** 4

**Summary:**

The choice of the prior distribution plays a central role in Bayesian inverse problems and a common way to select the prior is to use model evidence estimation. In recent years, diffusion models have emerged as powerful tools for constructing expressive priors across a range of challenging tasks. However, the corresponding model evidence is intractable in this framework as all marginal probability density functions are not available explicitly.

To address this limitation, the authors introduce DIME, a novel method for estimating the log-evidence of diffusion-based priors using only a limited number of samples. The proposed approach relies on samples drawn from the marginal posterior distributions of the diffusion process using a procedure inspired by Decoupled Annealing Posterior Sampling (DAPS). Furthermore, the authors enhance the Gaussian approximation of the denoiser by introducing a new covariance estimator, derived from a Gaussian approximation of the true underlying distribution.

The proposed estimator is evaluated using five baselines including sequential Monte Carlo methods across several scenarios: a high-dimensional Gaussian target with known ground truth, two non-convex inverse problems, and a model selection and validation task based on real observations of the M87* black hole. Empirical results demonstrate that DIME consistently outperforms all baseline approaches and effectively applies to real-world black hole imaging problems.

**Strengths:**

- The proposed method is derived using only a few posterior samples and the squared likelihood score can be estimated with state-of-the -art methods such as Decoupled Annealing Posterior Sampling.
- The algorithm is compared to various baselines in several challenging tasks and consistently outperforms the alternatives.
-  The model selection results for the real M87* data and the posterior samples with the lowest and highest evidence highlights that the method can be used in real data settings, using numerous potential priors.

**Weaknesses:**

- The proposed algorithm is well motivated and easy to derive from pretrained denoisers. However there are no theoretical guarantees or no intuitions on the data distribution for which the logevidence estimator is good.
- The algorithm uses DAPS whose annealing process reduces the dependency between samples at consecutive time steps,and allows to obtain marginal posterior samples for any noise level starting from samples from any other noise level. However, the reasearch on posterior sampling using pretrained score-based models has been very active for the past few years and alternatives could be considered. As there is no theoretical guarantee on the proposed estimate, exploring other samplers would support the choice made by the authors.
- The proposed estimate of the posterior covariance is based on a Gaussian approximation of the data distribution using training data. Although it provides a very easy to compute estimate, the assumption that the data distribution is close to a Gaussian is strong in settings where this distribution is likely to be multimodal.

**Questions:**

The covariance estimate proposed by Lemma 1 used a Gaussian assumption. Would you have insights on how to propose an estimate in a more general setting ? Or to provide guarantees that the proposed estimates is good even in non Gaussian settings.

The literature providing theoretical guarantees for score-based diffusion models is very rich for unconditional and conditional sampling. Could this theory be used to provide guarantees on the proposed logevidence estimator ?

Did you try score-based alternatives to DAPS to obtain the samples to support that this choice outperforms other posterior sampling algorithms based on pretrained diffusions ?

Could you include computational time to your experiments to understand the trade-off between number of samples from the diffusion and computational complexity ?

---

> ### Author Response · Authors · 2025-11-21
> **Response to Reviewer 56j4**
>
> Thank you for the comments and we are glad you think our method is “well motivated and easy to derive from pretrained denoisers”. Below we address your comments and questions:
>
> **W1**: There are a few conditions of the data and posterior distribution for the log evidence estimator to be good. The data and posterior distribution must be integrable for the proof to hold. The boundary conditions must also vanish for the integration by parts, and the distributions / corresponding vector fields must be smooth / continuous for Stein’s identity to hold. However, these same conditions are needed for training a diffusion model (see "Maximum Likelihood Training of Score-Based Diffusion Models"), and are almost always the case for real-world datasets. We will clarify this in the revised paper.
>
> **W2 & Q3**: Our method is reliant on an unbiased $\nabla \log p(y | x_t)$, as well as accurate posterior time-marginals. Therefore, guidance based methods (e.g., score-based alternatives that you are likely alluding to), which approximate $\nabla \log p(y | x_t)$, exhibit compounding bias and are not well suited for this task. For example, using the DPS approximation of $\nabla \log p(y | \mathbb{E}[x_0|x_t])$ leads to highly biased results: on the Mixture of Gaussians ablation study (Section 4.1), the resulting evidence estimate exceeds ~10000% relative error. We will revise the paper to add these results.
>
> Furthermore, the DAPS method was specifically chosen because not only is it the state-of-the-art posterior sampling method, it also generates pairs ($x_t \sim p(x_t|y)$, $x_0 \sim p(x_0|x_t,y)$) as a byproduct, which is required for an unbiased estimator of $\nabla \log p(y | x_t)$, making it the perfect fit for this method. We also discuss how to generalize our estimator to any posterior sampling method with known marginals in Appendix B.
>
>
> **W3 & Q1**: As mentioned in a previous comment for Reviewer w9CR, one can avoid using the Gaussian approximation by using the exact DAPS sampling, where the score of $p(x_0)$ is provided by querying the diffusion model at a tiny noise level. Empirically, both the original DAPS paper and us find that the Gaussian approximation leads to a similar quality of posterior samples and nearly identical evidence estimates, except for the RIAF prior which is extremely narrow and multimodal. We will revise the paper to explain this alternative more clearly and add in experiments that illustrate the tradeoff in accuracy/speed as a function of image size.
>
> |                          | **GRMHD prior** | **MNIST prior** | **Spacenet prior** | **CelebA prior** | **RIAF prior** |
> |:-------------------------|----------------:|----------------:|-------------------:|-----------------:|---------------:|
> | **DAPS (Gaussian, from Figure 3)** | $205 \pm 10$ | $82 \pm 27$ | $191 \pm 16$ | $170 \pm 24$ | $-212 \pm 154$ |
> | **DAPS (Exact)**         | $206 \pm 8$  | $75 \pm 18$ | $191 \pm 14$ | $182 \pm 8$  | $157 \pm 8$   |
>
>
> **Q2**: The primary guarantee for unconditional sampling that is needed for our estimator is that the estimated Tweedie mean by the network is equal to the true Tweedie mean $\mathbb{E}[x_0|x_t]$, as it is used to compute the integrand in the high noise estimator. Therefore, the theoretical guarantees will be identical to that of whether the network can correctly approximate the score of the implicit data distribution, and can be bounded by the weighted sum of score errors over all timesteps. We will revise the paper to add these results.
>
> **Q4**: All experiments were performed on a NVIDIA A100. For the black hole imaging case (64x64):
>
> DAPS (Gaussian approximation) & PnP-DM: 10 minutes for 20 posterior samples (i.e. 40 particles updated during langevin), 100 annealing steps, 1000 langevin steps.
>
> DAPS (Exact): 12 minutes for 20 posterior samples (i.e. 40 particles updated during langevin), 20 annealing steps, 1000 langevin steps.
>
> The computational speed is linear in the annealing steps and number of langevin steps, and approximately linear, but faster in practice due to parallelization, in the number of posterior samples.

---

> > ### Comment · Reviewer_56j4 · 2025-11-26
> >
> > I would like to thank the authors for their helpful answers and the proposed clarifications.
> >
> > I think that the additional experiment on the Gaussian approximation (which was also a concerned of Reviewer w9CR) and the use of DAPS sampling should be added in the article.

---

### Official Review · Reviewer_U7kJ · 2025-10-31

**Soundness:** 3
**Presentation:** 3
**Contribution:** 3
**Rating:** 6
**Confidence:** 4

**Summary:**

This paper introduces DiME, an estimator for the  marginal likelihood $\log p(y)$ when the prior is a pre-trained diffusion model. The key contribution is the estimation of a KL path integral representation of the evidence, leading to a practical estimator that leverages samples from the diffusion posterior sampling algorithm DAPS.

**Strengths:**

Through compelling applications with extensive experiments, the proposed method seems to exhibit strong performance. The authors have tested the method on:
- Controlled Gaussian prior benchmark:  here the method matches the performance of the SMC baseline.
- Phase retrieval on MNIST: DiME correctly identifies the generating prior for most digits; confusions appear only for symmetric cases and the method clearly outperforms SMC.
- Black hole imaging: Here the method provides results that align with physical evidence.

These results show that the method is in itself useful and could be interesting for the ICLR community.

**Weaknesses:**

I only see one weakness of this work:

- One of the contributions of the paper is the improved approximation for the posterior covariance but this approximation is already leveraged in a previous work that the paper doesn't cite [1]; see last paragraph of section 3 right before the algorithms. This is the same approximation as the one proposed in this work.

[1] Linhart, J., Cardoso, G.V., Gramfort, A., Corff, S.L. and Rodrigues, P.L., 2024. Diffusion posterior sampling for simulation-based inference in tall data settings.

**Questions:**

I am not sure to understand what's the difference between analytic score and clean score; cf line 309: the clean score is never used

---

> ### Author Response · Authors · 2025-11-21
> **Response to Reviewer U7kJ**
>
> Thank you for the comments and we are glad you think our method has “compelling applications with extensive experiments” and is “useful and could be interesting for the ICLR community”. Below we address your comments and questions:
>
> **W1**: Thank you for bringing the paper “Diffusion posterior sampling for simulation-based inference in tall data settings.” to our attention. We have added the corresponding citation.
>
> **Q1**: Analytic and clean score were both used in our paper to mean the same thing. Thank you for bringing up this confusion. We have removed the term “clean score” from our paper for consistency.

---

### Official Review · Reviewer_mdpM · 2025-11-01

**Soundness:** 3
**Presentation:** 2
**Contribution:** 3
**Rating:** 6
**Confidence:** 4

**Summary:**

This paper provides an efficient method for estimating normalising constants (model evidence) which can be utilized for model selection.

**Strengths:**

The paper presents the methodology cleanly and displays interesting empirical results, especially presenting an example on a GMM case as well as a practically interesting case of choosing priors for scientific datasets.

**Weaknesses:**

The paper has multiple weaknesses - ranging from the formulation and empirical results. Mainly

- Some of the theoretical results (see my comments) are well known and not novel
- Empirical comparisons miss some relevant work that also consider full covariance-based approximations of $p(x_0 | x_t)$

In general, it feels like the methodological contribution is limited, although I think experimental section is nicely done (see strengths)

**Questions:**

- The literature on alternative posterior covariance approximations were not surveyed properly, there are some relevant methods authors should discuss whether they can utilize it.

> Boys, B., Girolami, M., Pidstrigach, J., Reich, S., Mosca, A., & Akyildiz, O. D. Tweedie Moment Projected Diffusions for Inverse Problems. Transactions on Machine Learning Research, 2024.

Please discuss the approximation provided above in the light of Lemma 1. Would the use of approximations given above (or others in the literature) improve your method?

- Proposition 1 is written as a result of this paper, however, the formula for $\log p(y)$ is quite standard (just an application of Bayes rule) and the approximation of the KL divergence also follows from the definition of the diffusion model. I think this is misleading - authors should clarify that this is a standard and well-known consequence of Bayes rule.

- Can you explain the robustness properties of your method? If the data has outliers, how badly does it affect the results?

---

> ### Author Response · Authors · 2025-11-21
> **Response to Reviewer mdpM**
>
> Thank you for the comments, and we are glad you think the paper is “efficient” and “ presents the methodology cleanly and displays interesting empirical results,” Below we address your questions:
>
> **Q1**: While the above paper “Tweedie Moment Projected Diffusions for Inverse Problems” provides a closed-form posterior covariance, it can only be applied for linear inverse problems and not for general cases. However, we agree that it can be used to make evidence estimation of linear inverse problems more accurate under the Gaussian approximation, and will mention this in the revised paper.  Another useful approximation is parameterizing the covariance as diagonal in some sort of transform basis, as done in “Improving Diffusion Models for Inverse Problems Using Optimal Posterior Covariance”.
>
> Furthermore, as described in a comment for Reviewer w9CR, the Gaussian covariance issue can be sidestepped using the exact DAPS sampling, where the score of $p(x_0)$ is provided by querying the diffusion model at a tiny noise level. Both the original DAPS paper and we find that the Gaussian approximation leads to a similar quality of posterior samples and nearly identical evidence estimates for all the real-world priors used in the black hole imaging section, except for the RIAF prior which is extremely narrow and multimodal.
> |                          | **GRMHD prior** | **MNIST prior** | **Spacenet prior** | **CelebA prior** | **RIAF prior** |
> |:-------------------------|----------------:|----------------:|-------------------:|-----------------:|---------------:|
> | **DAPS (Gaussian, from Figure 3)** | $205 \pm 10$ | $82 \pm 27$ | $191 \pm 16$ | $170 \pm 24$ | $-212 \pm 154$ |
> | **DAPS (Exact)**         | $206 \pm 8$  | $75 \pm 18$ | $191 \pm 14$ | $182 \pm 8$  | $157 \pm 8$   |
>
> **Q2**: It was not our intention to make the formula in Proposition 1 sound like a novel result, but rather a proposed approach for our problem. We will clarify this. To our knowledge, this KL integration approach has not been used before to estimate the model evidence for diffusion models.
>
> **Q3**: Our results demonstrate minimal error when the measurement $y$ is out-of-distribution in the Mixture of Gaussian setting, but this setting assumes a perfect score function with no errors. We also perform out-of-distribution experiments for both the MNIST example and in Appendix F: while there is no way to verify that the estimated evidence of out-of-distribution measurements is exact, we find that the correct model is chosen in almost all cases, and incorrect models are always assigned relatively lower evidence.

---

### Official Review · Reviewer_w9CR · 2025-11-02

**Soundness:** 2
**Presentation:** 2
**Contribution:** 2
**Rating:** 2
**Confidence:** 4

**Summary:**

- The paper addresses the problem of selecting an appropriate prior model in inverse problems by proposing an estimator of the model evidence
- The estimator leverages samples generated during the diffusion reverse process and decomposes the evidence computation into two components: the expected log-likelihood and KL divergence whose considered approximation involves the norm of the gradient of the intermediate log-likelihood
- The intermediate likelihoods are approximated using two estimators depending on low/high noise setting.
- As an example, the authors apply the approach to the DAPS and PnP-DM samplers
- for DAPS, they also introduce a new approximation of the covariance $X_0 | X_t$, estimated empirically from the training dataset through the covariance of the prior $p_0$

**Strengths:**

- The introduced procedure to estimate the model evidence is practical and may enable informed model selection in inverse problems.
- the proposed approximation of the conditional covariance $X_0 | X_t$ within the DAPS framework is novel

**Weaknesses:**

**Methodological issues**
- The authors criticize alternative methods for model evidence approximation for being inaccurate and biased (Line 52-53 and Line 64-65). this claim is misleading as the proposed approach relies heavily on Gaussian approximations; and more precisely, the authors don't quantify the impact of such assumption on the approximation
- In Lemma 2, the variance computation of both estimators is incorrect. In particular, equation (12) incorrectly states that $Var(\Theta_{\text{high}}) = O(1/\sigma_t^2)$; however, this ignores the dependence of $\Sigma_{0|t}$ on $\sigma_t^2$, which would yield, if considered, a variance $O(1)$. Consequently, the practical criterion for choosing between low- and high-noise estimators is based on a flawed derivation. In addition, in practice, the authors choose the variant of the estimator with the lowest variance, but this irrelevant as the variance of the estimator are computed with different asymptotic assumptions
- It is concerning that the high-noise estimator in equation (11) does not depend on the observation $y$, but may still be used during the algorithm
- The proposed approximation of $Cov(X_0 \mid X_t)$ is impractical. the authors propose to estimate the empirical covariance $\Sigma_0$ of $p(x_0)$ from training data but this computationally expensive (quadratic memory cost); in addition, this covariance $\Sigma_0$ is involved in matrix-matrix operations and must also be inverted

**Typos and technical inaccuracies**
- Section 2.2: The categorization of posterior sampling methods is irrelevant and doesn't reflect state-of-the-art. Methods such as TDS [2] and MCGDiff [3] are based on SMC and evolve particle ensembles to solve inverse problems; MGPS [4] uses midpoint guidance for estimating transitions; and RedDiff [5] is a fully variational approach.
- Line 269: The equation is mathematically incorrect: the square of $\Theta$ is computed but $\Theta$ is a vector.
- Lines 60–63: The claim that computing the log-density of a prior sample requires integrating the Jacobian of the score overlooks existing work (see Eq. (13) in [1]), where this can be done without vector jacobian product over the score.
- The statement in Lines 207–209 is ambiguous, at low noise, $p(x_t \mid x_0)$ concentrates near $x_0$, so saying it "dominates over the Gaussian approximation of $p(x_0)$" lacks clear meaning.

**Limitations**
The method is applicable in settings where the posterior sampler follows a noise–then-denoise procedure.
This restriction should be explicitly stated. Furthermore, the paper does not discuss how $\Sigma_{0|t}$ is handled in practice. Since the space complexity of thus matrix scales quadratically with the dimension if the problem and appears in the estimator’s computation, it poses computational/memory challenges


---

.. [1] Skreta, Marta, et al. "The Superposition of Diffusion Models Using the It\^ o Density Estimator." arXiv preprint arXiv:2412.17762 (2024).

.. [2] Wu, Luhuan, et al. "Practical and asymptotically exact conditional sampling in diffusion models." Advances in Neural Information Processing Systems 36 (2023): 31372-31403.

.. [3] Cardoso, Gabriel, et al. "Monte Carlo guided diffusion for Bayesian linear inverse problems." arXiv preprint arXiv:2308.07983 (2023).

.. [4] Moufad, Badr, et al. "Variational diffusion posterior sampling with midpoint guidance." arXiv preprint arXiv:2410.09945 (2024).

.. [5] Mardani, Morteza, et al. "A variational perspective on solving inverse problems with diffusion models." arXiv preprint arXiv:2305.04391 (2023).

**Questions:**

- Since Equation 9 intervenes computation of quantities under different the expectation of different marginals: how many sample paths are used to evaluate them in Algorithm 1 ?
- Can the authors provide the runtime of the algorithms (For DAPS and PnP-DM)?

---

> ### Author Response · Authors · 2025-11-21
> **Response to Reviewer w9CR**
>
> Thank you for your detailed and constructive comments. We are glad you believe our method is “practical and may enable informed model selection in inverse problems”. Below we address your questions and comments, most of which stem from misunderstanding that can be easily clarified in the text:
>
> **W1, W4 & Limitation**: While our approach does rely on Gaussian approximations, our aim in the paper is to introduce a general evidence estimation framework for diffusion model priors along with one practical implementation that uses this approximation. As shown in Table 1 in our paper, our proposed approach produces evidence estimates that closely match the ground truth for multimodal priors. We also discuss how to extend this framework to other samplers in Appendix B that may not rely on this approximation.
>
> We agree that storing a full covariance matrix becomes impractical for very high image dimensions.  For this case, and to also avoid the Gaussian approximation, the original DAPS paper also describes an exact posterior-sampling alternative that decomposes the score of $p(x_0∣x_t)$ into the score of $p(x_0)$ (obtained by setting a very low noise level in the diffusion model) and $p(x_t|x_0)$ (Gaussian)​. This can handle large images and removes the error incurred by the Gaussian approximation. Below we show the evidence computed using both methods for the black hole model selection problem. The Gaussian approximation produces evidence estimates that closely match that of the exact posterior sampling approach across a variety of real-world data-driven priors. The RIAF prior is the only outlier, where the exact sampler’s lower-variance estimate reveals that the Gaussian approximation degrades for unusually narrow and multimodal priors. We will revise the paper to explain this alternative more clearly and add in experiments that illustrate the tradeoff in accuracy/speed as a function of image size.
>
> |                          | **GRMHD prior** | **MNIST prior** | **Spacenet prior** | **CelebA prior** | **RIAF prior** |
> |:-------------------------|----------------:|----------------:|-------------------:|-----------------:|---------------:|
> | **DAPS (Gaussian, from Figure 3)** | $205 \pm 10$ | $82 \pm 27$ | $191 \pm 16$ | $170 \pm 24$ | $-212 \pm 154$ |
> | **DAPS (Exact)**         | $206 \pm 8$  | $75 \pm 18$ | $191 \pm 14$ | $182 \pm 8$  | $157 \pm 8$   |
>
> **W2**: Thank you for raising this point of confusion. The full variance derivation, including the dependence on $\Sigma_{0|t}$ and the observation term, appears in Appendix A.1, and we will add an explicit reference in the main text to make this clearer. In that derivation, the $\Sigma_{0|t}$ dependence is absorbed into the final bound. Both estimators also display their expected variance properties empirically.
>
> **W3**: This is addressed in the appendix as well. The bound shown in the lemma reflects the worst-case scenario, in which the data y is extremely weak so $Cov(x_0 | x_t, y)$ is approximately $Cov(x_0 | x_t)$.
>
> **TTI1**: We will add the references to TDS, MCDiff, MGPS and RedDiff to the introduction, and extend the previous “proximal methods” categorization to methods that anneal along specific intermediate marginals. Our approach can indeed be adapted to TDS and MGPS, since they use the closed-form marginals $p(x_t) p(y | \mathbb{E}[x_0|x_t])$. However, as shown in “InverseBench: Benchmarking Plug-and-Play Diffusion Priors for Inverse Problems in Physical Sciences,” DAPS is a state-of-the-art plug-and-play diffusion method that is a top performer across a broad range of inverse problems, and in fact outperforms RedDiff and MCGDiff on several of the tasks we study in our paper. Furthermore, MCGDiff is limited to linear inverse problems, and RedDiff performs variational MAP estimation rather than multimodal posterior sampling, making it unsuitable for Bayesian evidence estimation.
>
> **TTI2 & TTI4**: Thank you for pointing this out. We corrected the typo in the equation and revised the statement in Lines 207-209 to remove the ambiguous wording.
>
> **TTI3**: Thank you for bringing this work to our attention. We missed it during our literature review and will add it to the paper. However, density-based methods for evidence estimation still require numerous diffusion samples, making them infeasible for diffusion-based methods.
>
> Limitation: While the two sampling methods discussed in our paper are noise-then-denoise, any diffusion posterior sampling method that anneals along specified marginal distributions is a candidate for our evidence estimator, as shown in Appendix B. We will clarify this weaker restriction in the paper. Furthermore, most noise–then-denoise posterior samplers can be applied to any pretrained diffusion model, making this requirement not very restrictive in practice.
>
> **Q1**: The number of sample paths (posterior samples) can be any amount, and we used 20 in all of our experiments.
>
> **Q2**: See Q4 for reviewer 56j4.

---

### Author Response · Authors · 2025-12-03
**Summary of rebuttal**

We thank all reviewers for their constructive and thoughtful feedback, and we appreciate the AC’s careful attention to the discussion. We believe all concerns arise either from misunderstandings that can be easily clarified in the revision or from points that require only minor additions, all of which we have already addressed in our discussion. We also wish to note for the AC that Reviewer 56j4 had raised their score to a 6 on Nov 26 before the comments were reverted. The other reviewers did not get a chance to respond to our rebuttal.

We summarize the key points raised by reviewers and our responses as well as additional results we will add to the paper:

**The Gaussian approximation / covariance estimation**

Multiple reviewers raised concerns about the Gaussian approximation of $p(x_0 | x_t)$: namely, that approximating $p(x_0 | x_t)$ as Gaussian would be inaccurate for real-world multimodal, complex distributions, and that storing an empirical covariance matrix for larger images would be computationally infeasible. To simultaneously address both of these issues, we also demonstrate that DiME can be used with the exact DAPS posterior sampling method, which uses the diffusion model output at a tiny noise level to estimate the true score of $p(x_0)$. Our results on the black hole imaging case demonstrate that the exact formulation leads to highly similar and lower variance evidence estimates compared to the Gaussian approximation with only about a 20% computational slowdown. This approach also avoids having to compute any sort of covariance matrix or approximation. Reviewer 56j4 (the only reviewer who had a chance to respond) was happy with this addition and asked us to add it to the paper.  We will further explore the tradeoffs in a revised version of the work.

|                          | **GRMHD prior** | **MNIST prior** | **Spacenet prior** | **CelebA prior** | **RIAF prior** |
|:-------------------------|----------------:|----------------:|-------------------:|-----------------:|---------------:|
| **DAPS (Gaussian, from Figure 3)** | $205 \pm 10$ | $82 \pm 27$ | $191 \pm 16$ | $170 \pm 24$ | $-212 \pm 154$ |
| **DAPS (Exact)**         | $206 \pm 8$  | $75 \pm 18$ | $191 \pm 14$ | $182 \pm 8$  | $157 \pm 8$   |


**Applicability to other Diffusion Posterior Sampling methods**

Reviewer w9CR also raised concerns about the generalizability of our method. While we only demonstrate our estimator on two variable splitting methods, DAPS and PnP-DM, we can extend it to any diffusion posterior sampling method that relies on analytic marginals. For example, the methods MGPS and TDS were brought up which do not require noise-then-denoise sampling and also adhere to the intermediate marginals $\\pi(x_t | y) \propto p(x_t) p(y | E[x_0 | x_t])$. We can do a similar derivation to the PnP-DM marginals to obtain the evidence gradient for MGPS / TDS marginals:

$\\frac{d}{dt} \\log \\pi(y \\mid t)$
 = $E_{x_t \sim \pi(x_t \mid y)}\left[v_p(x_t)^T \nabla_{x_t} \log p(y \mid \mathbb{E}[x_0 \mid x_t]) + \frac{d}{dt} \log p(y \mid \mathbb{E}[x_0 \mid x_t])\right].$

We will also include this derivation and detailed comparison experiments in the paper’s supplementary material. Notably, this estimator only requires the PF-ODE velocity field (obtained from the diffusion model), the likelihood term value and its time-derivative, which can easily be obtained via autodifferentiation.

---

### Public Comment · ~Frederic_Wang1 · 2026-02-25
**Issue with black hole likelihood**

We discovered a bug with the computation of the black hole likelihood in the InverseBench library, where it was incorrectly multiplied by an extra factor of 2. We have fixed the issue and re-ran all of the black hole experiments; as a result, all computed evidences have shifted upward roughly uniformly, but all of our scientific conclusions still remain the same. We apologize if this caused any confusion.

---

### Meta-Review · Area_Chair_F1Xy · 2026-01-06

**Summary:**

This paper tackles principled prior selection in diffusion-based inverse problems. It proposes a practical estimator of model evidence using samples from the diffusion reverse process. The evidence is decomposed into an expected log-likelihood term and a KL term. The KL component is approximated efficiently using the norm of the gradient of intermediate log-likelihood. The intermediate likelihood is estimated with two schemes for low- and high-noise regimes. The approach is demonstrated on DAPS and PnP-DM, and DAPS additionally uses an empirical covariance approximation from training data. The method is presented clearly and evaluated convincingly. In MNIST phase retrieval, DiME identifies the generating prior for most digits and outperforms SMC, with errors mainly in symmetric cases. In black hole imaging, the selected priors are consistent with physical evidence. Reviewer w9CR's concerns about mathematical issues were addressed in the revision. Overall, I find the contribution solid and recommend acceptance.

**Reviewer Concerns:**

.

**Reviewer Scores:**

.

---

### Decision · Program_Chairs · 2026-01-26

Accept (Poster)